# Sequence-encoded and composition-dependent protein-RNA interactions control multiphasic condensate morphologies

Taranpreet Kaur[1], Muralikrishna Raju [2,3], Ibraheem Alshareedah[1,3], Richoo B. Davis [1], Davit A. Potoyan[2] & Priya R. Banerjee [1]

Multivalent protein-protein and protein-RNA interactions are the drivers of biological phase separation. Biomolecular condensates typically contain a dense network of multiple proteins and RNAs, and their competing molecular interactions play key roles in regulating the condensate composition and structure. Employing a ternary system comprising of a prion-like polypeptide (PLP), arginine-rich polypeptide (RRP), and RNA, we show that competition between the PLP and RNA for a single shared partner, the RRP, leads to RNA-induced demixing of PLP-RRP condensates into stable coexisting phases—homotypic PLP condensates and heterotypic RRP-RNA condensates. The morphology of these biphasic condensates (non-engulfing/ partial engulfing/ complete engulfing) is determined by the RNA-to-RRP stoichiometry and the hierarchy of intermolecular interactions, providing a glimpse of the broad range of multiphasic patterns that are accessible to these condensates. Our findings provide a minimal set of physical rules that govern the composition and spatial organization of multicomponent and multiphasic biomolecular condensates.

[1] Department of Physics, University at Buffalo, Buffalo, NY, USA. [2] Department of Chemistry, Iowa State University, Ames, IA, USA. [3] These authors contributed equally: Muralikrishna Raju, Ibraheem Alshareedah. ✉email: potoyan@iastate.edu; prbanerj@buffalo.edu

In biological cells, many multivalent ribonucleoproteins (RNPs) form biomolecular condensates that act as active or repressive hubs for intracellular storage and signaling[1,2]. These condensates can rapidly assemble and dissolve in response to cellular stimuli via a physical process known as liquid–liquid phase separation[3–5]. The functional specificity of biomolecular condensates as subcellular organelles is linked to selective enrichment of specific enzymes/signaling factors within[1,6], whereas altered compositions of signaling condensates are associated with disease pathologies[2,4,7–9]. Mounting evidence now suggests that the spatial organization of biomolecules into distinct sub-compartments within biomolecular condensates [e.g., nuclei[10,11], nuclear speckles[12], paraspeckles[13], and stress granules[14]] adds another layer of internal regulation of composition and plays a fundamental role in facilitating their complex biological functions. These mesoscopic multilayered structures can be qualitatively understood based on a multi-phasic condensate model, where two or more distinct types of partially immiscible condensed phases are formed by spontaneous phase separation of individual components in a multi-component mixture[11,15,16]. In this work, we set out to study the underlying molecular mechanisms that regulate the multiphasic condensate composition and spatial organization by employing a tractable, minimalistic ternary system.

The composition and spatial organization of biomolecular condensates are ultimately controlled by the nature of intermolecular interactions between RNPs and RNAs as well as their interactions with the solvent molecules[2,11,17,18]. Analysis of sequence features of eukaryotic biomolecular condensate proteins revealed that intrinsically disordered low-complexity domains (LCDs) are common drivers and/or regulators of RNP phase separation with and without RNAs[19–23]. The LCD sequence composition and patterning provide programmable modules for dynamic multivalent protein–protein and protein–RNA interactions[24–29]. In multi-component mixtures, these dynamic inter-chain interactions are ubiquitous and can either cooperate or compete within a dense network of LC proteins and RNAs[30,31]. The presence of multiple RNPs and RNAs within an intracellular biomolecular condensate highlights the relevance of understanding how networks of competing interactions control the condensate composition and structure[32], given these very properties are intricately linked to their functional output in the cell[2].

To systematically explore the regulatory principles of multi-component RNP condensation with competing protein–protein and protein–RNA interactions, here we employ a minimalistic three-component system composed of two LC disordered polypeptides: a prion-like polypeptide (PLP) and an Arg-rich polypeptide (RRP), and RNA. PLPs are typically characterized by the presence of π electron-rich and polar amino acids[33,34] (Y/N/Q/G/S; examples: hnRNPA1, TDP43, FUS)[23], whereas R-rich polypeptides bind RNAs with a broad range of sequence composition and structures[35], and commonly occur as intrinsically disordered RGG domains[22,36] (examples: G3BP1, LSM14A, hnRNPDL, EWSR1, FUS, TAF15). Both LCD types are highly abundant in stress granule and processing body proteins[37]—the two major cytoplasmic biomolecular condensates in eukaryotes. From a pathological point of view, multivalent R-rich repeat polypeptides, such as poly(GR) and poly(PR), are potent neurotoxins and are directly linked to c9orf72-derived repeat expansion disorder[38–44]. These R-rich repeat polymers can invade SGs and impair their fluid dynamics by aberrantly interacting with SG components, including PLPs and RNAs[29,45–47]. Within the PLP-RRP-RNA system, three interactions have the capacity to drive independent phase separation processes: PLP-PLP interactions can drive homotypic PLP phase separation[23,48], PLP-RRP interactions can drive the co-phase separation of PLP-RRP into

heterotypic condensates[26], and RRP-RNA interactions can drive the formation of RRP-RNA condensates[28,49,50]. Therefore, the RRP within the PLP-RRP-RNA system represents a common module that can interact with both PLPs and RNAs[26,45,46]. As such, the PLP-RRP-RNA ternary system represents a suitable and biologically relevant triad for dissecting how networks of competing biomolecular interactions control the condensate composition and structure.

To systematically study the role of competitive protein–protein and protein–RNA interactions in controlling the organization of ternary PLP-RRP-RNA condensates, here we employ a multi-scale biophysical approach. For two-component systems (PLP-RRP and RRP-RNA), we show that the mixture composition is a key factor in controlling the phase behavior, condensate spatial organization, and client recruitment in a context-dependent manner. Specifically, we show that for PLP-RRP mixtures, RRP monotonically enhances PLP condensation. On the contrary, RRP-RNA mixtures display a reentrant phase behavior in which the RNA-to-RRP mixing ratio determines the surface organization of RRP-RNA binary condensates in a non-monotonic fashion. Within the PLP-RRP-RNA ternary system, RNA-RRP interactions dominate over PLP-RRP interactions, leading to an RNA-induced demixing of PLP-RRP condensates into stable coexisting phases (homotypic PLP condensates and heterotypic RRP-RNA condensates). The organization of these biphasic condensates (non-engulfing/partial engulfing/complete engulfing) is determined by the RNA-to-RRP stoichiometry and the hierarchy of intermolecular interactions, providing a glimpse of the broad range of multiphasic patterns that are accessible to these condensates. Mechanistically, the multiphasic structuring of PLP-RRP-RNA condensates is governed by the molecular interactions at the liquid-liquid interface, which are encoded in the amino-acid sequence of the proteins and regulated by the composition of the mixture. This multi-scale regulation of inter-condensate surface interactions controls the relative interfacial tensions between the three liquid phases (PLP condensed phase, RRP-RNA condensed phase, and the dispersed phase). Together, our findings reveal that competing intermolecular interactions in a multicomponent system represent a regulatory force for controlling the composition and structure of multiphasic biomolecular condensates.

## Results

**Mixture composition controls the structure and dynamics of binary condensates.** Before probing the ternary PLP-RRP-RNA phase separation, we studied the phase behavior of the three corresponding binary mixtures: PLP-RRP; PLP-RNA; and RRP-RNA. The presence of R-rich RNA-binding domains has been reported to enhance PLP phase separation through intermolecular Arg-Tyr interactions[26,45]. To quantify such an effect in our PLP-RRP binary system, we first determined the isothermal state diagram for $FUS^{PLP}$ in the presence of arginine/glycine-rich polypeptides (RRP: $[RGRGG]_5$ and $FUS^{RGG3}$; Table S2). Our analysis revealed that the phase separation of PLP is enhanced with RRP in a composition-dependent manner. Specifically, the LLPS concentration threshold for PLP ($C_{LLPS}^{PLP}$) decreases monotonically with increasing RRP concentration [in the absence of the RRP, $C_{LLPS}^{PLP} = C_{saturation}^{PLP} \sim 240\,\mu M$; at an RRP-to-PLP ratio of 5, $C_{LLPS}^{PLP} \sim 120\,\mu M$] (Fig. 1a; Figs. S1 and S2) under our experimentally tested conditions (up to RRP-to-PLP ratio of 20:1). Although PLP readily undergoes homotypic condensation in the absence of RRP, the latter did not show any sign of homotypic LLPS at the concentrations used in our study. Hence, our results that the saturation concentration of PLP decreases monotonically with [RRP] are consistent with a scenario where RRP acts as a

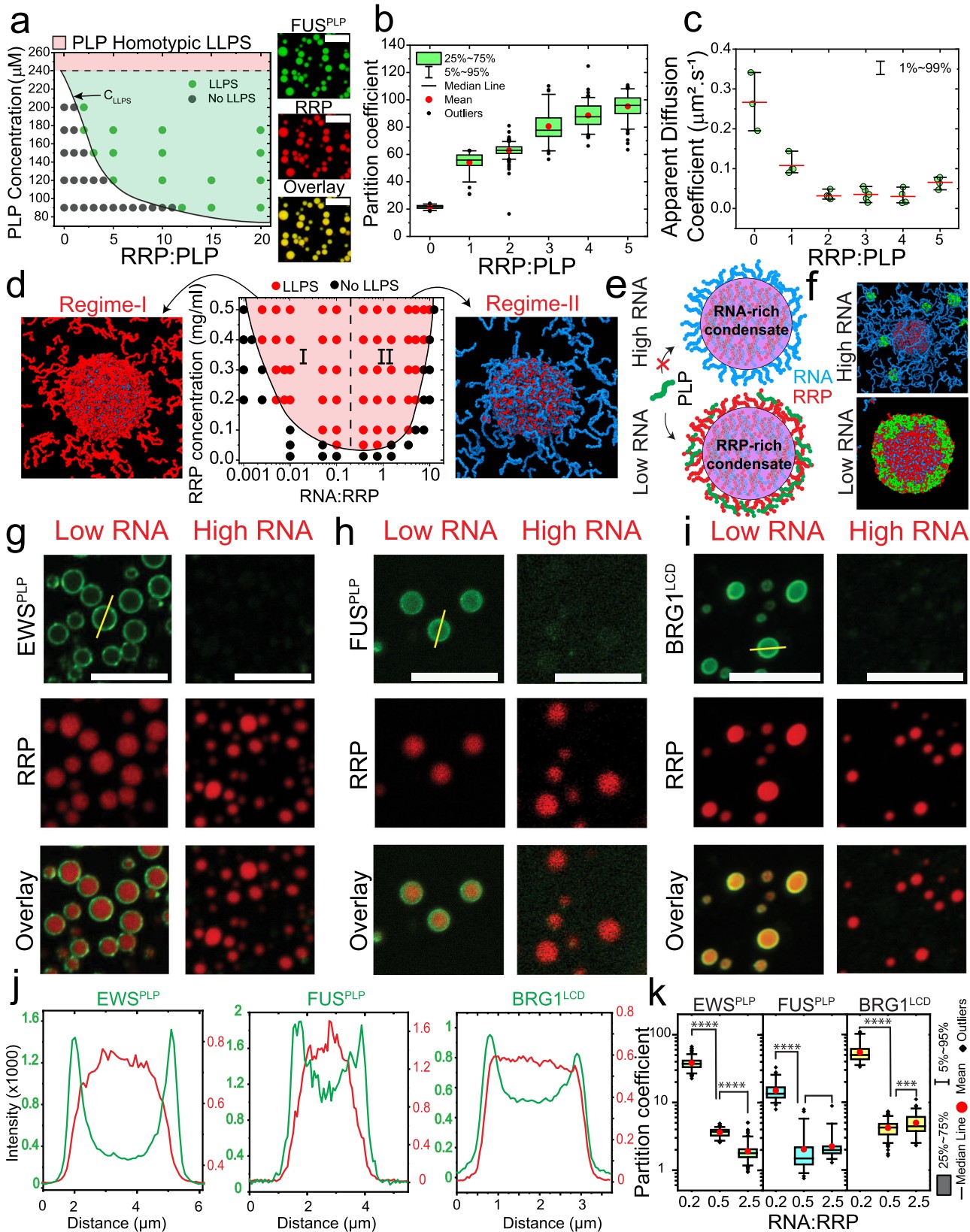

ligand that binds to the PLP preferentially in the dense phase[51,52]. In addition to the altered phase behavior of PLP due to the presence of RRP, we observe that the PLP partition coefficient ($K = C_{Dense}^{PLP} / C_{Dilute}^{PLP}$) increases with [RRP]. This is a direct manifestation of the lowering of PLP saturation concentration by the RRP. Simultaneously, fluorescence recovery after photobleaching (FRAP) assays indicate that the PLP mobility ($D_{app}$) in the dense phase decreases with increasing [RRP] (Fig. 1b and c,

**Fig. 1 Mixture composition controls the structure and dynamics of binary condensates. a** Left: State diagram of PLP-RRP mixtures with RRP-to-PLP mixing ratio (mole: mole). Shaded green region: co-phase-separation regime for PLP-RRP (PLP homotypic saturation concentration: $C_{sat} = 240\ \mu M$, RRP= [RGRGG]$_5$). Shaded regions are drawn as a guide to the eye. Right: Representative fluorescence images of PLP-RRP condensates. Scale bars, 20 μm. **b** PLP partition ($n = 60$ droplets per sample), and **c** PLP diffusion rate ($n = 3$ droplets per sample) in PLP-RRP condensates at variable RRP-to-PLP mixing ratios (mole/mole). The error bars are defined as the range of the data (1–99%) while the red line represents the mean value. **d** Center: State diagram of RRP-RNA mixtures with RNA-to-RRP mixing ratio (wt/wt). The shaded region shows the phase-separation regime and is drawn as a guide to the eye. Equilibrium MD configurations of condensates at $C_{RNA} = 0.5 \times C_{RRP}$ (left), and $C_{RNA} = 1.7 \times C_{RRP}$ (right). RRP: red; RNA: blue. $C_{RRP} = 1.3$ mg/ml. **e** A schematic diagram showing that RRP decorates the RRP-RNA condensates' surface at $C_{RRP} > C_{RNA}$ while RNA surface enrichment occurs at $C_{RNA} > C_{RRP}$, leading to differential surface recruitment of PLP clients (green) in the two types of condensates. **f** Equilibrium MD configurations of RRP-RNA condensates with PLP clients (green) at low-RNA and high-RNA ($C_{RNA} = 0.5 \times C_{RRP}$ and $C_{RNA} = 1.7 \times C_{RRP}$, respectively). RRP: red; RNA: blue. $C_{RRP} = 1.3$ mg/ml, $C_{PLP} = 0.4$ mg/ml. **g–i** Fluorescence microscopy images showing the recruitment behavior of EWS$^{PLP}$, FUS$^{PLP}$, and BRG1$^{LCD}$ into RRP-RNA droplets at variable RNA-to-RRP ratios. Scale bars represent 10 μm. **j** Intensity profiles across RRP-RNA condensates at low-RNA concentration (yellow lines in **g–i**) showing the surface recruitment of EWS$^{PLP}$, FUS$^{PLP}$, BRG1$^{LCD}$ (green: client, red: RRP). **k** Client partition coefficient in RRP-RNA condensates at variable RNA-to-RRP mixing ratios for FUS$^{PLP}$ ($n = 50$ droplets), EWS$^{PLP}$ ($n = 100$ droplets), and BRG1$^{LCD}$ ($n = 75$ droplets). A two-tailed $t$-test with no adjustments was used for statistical analysis (****$p$-value < 0.0001, ***$0.0001 < p$-value < 0.001 and no star: $p$-value > 0.05). See source data file for source data (**b–c**, **k**) and $p$-values. For **g–k**, samples were prepared at [FUS$^{RGG3}$] = 1 mg/ml and at a poly(rU)-to-FUS$^{RGG3}$ weight ratio of 0.2 (low RNA) and 2.5 (high RNA) or as indicated. For **d** and **f**, RRP = FUS$^{RGG3}$; RNA = poly(rU). For **a–c**, **f**, PLP = FUS$^{PLP}$. Buffer: 25 mM Tris-HCl (pH = 7.5), 150 mM NaCl, 20 mM DTT.

Figs. S3 and S4). These results are consistent with previous literature reports[26,45], and indicate that RRPs prefer binding to the PLP in the dense phase[51,52], thereby enhancing PLP phase separation and impacting the condensate dynamics (Fig. 1a–c). However, in contrast to PLP-RRP mixtures, the isothermal state diagram of PLP-RNA mixtures [utilizing a homopolymeric RNA, poly(rU)] showed that RNA does not have any significant impact on PLP phase separation (Fig. S5a). We independently confirmed that poly(rU) RNA does not significantly interact with the PLP by using fluorescence correlation spectroscopy (FCS), which revealed identical PLP autocorrelation curves in the absence and presence of poly(rU) RNA (Fig. S5b) in the single-phase regime. Furthermore, partition analysis of an RNA oligomer (rU10) showed a partition coefficient of ≤1.0 in PLP droplets (Fig. S6). Combining the state diagram, FCS, and partition analyses, we conclude that poly(rU) RNA does not have significant interactions with the PLP.

Similar to RRP-PLP mixtures, RRP-RNA mixtures display a composition-dependent phase behavior. However, unlike RRP-PLP mixtures, their phase behavior is non-monotonic, wherein the two-phase regime is only stabilized within a small window of mixture compositions (Fig. 1d). Such composition-dependent phase separation is a hallmark of multi-component systems and is usually referred to as reentrant phase transition[28,49,53]. The observed difference in the phase behavior of RRP-RNA and RRP-PLP mixtures is expected in light of recent theoretical developments which indicate that multicomponent mixtures with obligate heterotypic interactions often display reentrant phase behavior[54–60]. Within the reentrant phase separation window (Figs. 1d and S7), we previously predicted that disproportionate mixture compositions may lead to the formation of spatially organized condensates, in which the condensates' surfaces can be either enriched in RRPs or RNAs depending on the mixture stoichiometry[53]. In order to provide a molecular-level understanding of these assemblies, we performed molecular dynamics (MD) simulations using a single-residue resolution coarse-grained model of an RRP from an archetypal ribonucleoprotein FUS (FUS$^{RGG3}$) and a homopolymeric RNA, poly(rU)[53] (Table S2). The representative equilibrium structures of the RRP-RNA condensates indeed revealed two distinct condensate architectures: (a) condensates with RRP-enriched surfaces at $C_{RNA} < C_{RRP}$, and (b) condensates with RNA-enriched surfaces at $C_{RNA} > C_{RRP}$ (Figs. 1d and S8). Therefore, these condensates appear to be spatially organized[53] and their surface composition is dynamically varied in an RNA dose-dependent manner.

Since RRP chains, but not RNA chains, multivalently interact with the PLP chains (Figs. 1a–c and S5 and S6), we next considered the potential of RRP-RNA condensates to differentially recruit prion-like clients based on their surface architecture. We hypothesize that RRP-rich condensates, but not the RNA-rich condensates, would positively recruit PLPs. According to our proposed model (Fig. 1d), the recruited PLPs would be preferentially localized on the surface of RRP-rich condensates due to the availability of free RRP sites (Fig. 1e). MD simulations similar to those in Fig. 1d but now including π-rich FUS$^{PLP}$ as a client (Table S2) revealed enhanced surface recruitment of PLP chains into RRP-RNA droplets only at low-RNA conditions (Fig. 1f, Fig. S9), thereby lending support to this idea. To test this experimentally, we utilized PLPs from two different RNPs, EWS and FUS, and quantified their partitioning in FUS$^{RGG3}$-poly(rU) condensates (Table S2). Briefly, we formed RNA-RRP condensates at variable RNA-to-RRP mixing ratios in a buffer that contains the desired PLP clients (labeled with Alexa488 dye). Confocal fluorescence microscopy assays showed that both PLPs are preferentially recruited into RRP-RNA condensates at $C_{RRP} > C_{RNA}$ while the same clients showed no preferential partitioning into the RRP-RNA condensates at $C_{RRP} < C_{RNA}$ (Fig. 1g, h, Figs. S10, S11). Analysis of these fluorescence micrographs reveals ~10-fold increase in the partition coefficient of PLPs in RRP-rich condensates as compared to RNA-rich condensates (Fig. 1g, h, k). Furthermore, inspecting the fluorescence intensity profiles across RRP-rich condensates revealed that PLPs are preferentially recruited on the condensates' surface while being relatively depleted from the condensates' core (Fig. 1j). These observations confirm that PLP recruitment in RRP-rich condensates is mediated by molecules on the surface which are predominantly, as our simulation suggests, unbound segments of RRPs (Fig. 1d). To test the generality of this phenomenon, we next performed a similar analysis with two additional client polypeptides with similar sequence features as FUS and EWS PLPs: the P/S/G/Q-rich LCD of a transcription activator BRG1 (AA: 1–340) and the C-terminal LCD of RNA polymerase II (Pol II CTD) which contains 30 repeats of YSPTSPS (Table S2). Both BRG1$^{LCD}$ and Pol II CTD are enriched in amino acid residues which have exposed π-containing peptide backbones[21]. Besides, Pol II CTD is also enriched in Tyrosine residues (Table S2). Thus, both BRG1$^{LCD}$ and Pol II CTD are expected to interact with RRPs through Arg-π contacts and therefore display RNA-dependent recruitment into spatially organized RRP-RNA condensates. This prediction was verified

experimentally using our confocal microscopy assay (Fig. 1i–k, Figs. S12, S13). Although the magnitude of relative surface enrichment of PLP and π-rich clients seem to vary with the client used (Fig. S14), the existence of such surface enrichment is general to the tested client proteins. Such system specificity may arise from the varying interaction strength between RRP and the different client proteins. We further confirm that this surface enrichment is not specific to RNA by repeating the same assay with RRP-poly(phosphate) condensates (Fig. S15). We note that our results of PLP client recruitment preferentially on the surface of RRP-RNA condensates bears similarity to a recent report of enhanced surface localization of several fluorescently labeled mRNAs to RNA-only condensates in vitro formed by poly(rA) RNA as well as RNP condensates such as purified stress granules from mammalian cells[61]. Similar surface localization was also observed in MD simulations of condensates formed by Arg-rich disordered proteins and polynucleotides[62].

Collectively, our analysis of the phase separation behavior of different binaries in the PLP-RRP-RNA ternary system reveals two significant pairwise interactions: PLP-RRP and RRP-RNA. While the mixture composition in the PLP-RRP system monotonically impacts PLP-RRP condensate dynamics, in the RRP-RNA binary system, it controls the RRP-RNA condensate architecture. The stoichiometry-dependent recruitment of PLP clients and their spatial localization within RRP-RNA condensates highlights that RNA is capable of regulating PLP-RRP interactions by controlling the availability of free RRP chains on the surface of these condensates (Fig. 1e). As such, the phase behavior and compositional control of the PLP-RRP-RNA ternary system are expected to be governed primarily through heterotypic interactions between RRP and RNA. To test this idea, we next examined the impact of RNA on the phase behavior and organization of PLP-RRP condensates.

**RNA induces condensate switching from PLP-RRP to RRP-RNA droplets**. To probe for the effect of RNA on the heterotypic PLP-RRP condensates, we first generated PLP-RRP condensates at a PLP concentration lower than the homotypic PLP LLPS concentration ($C_{PLP} < C_{sat}$; the green region in the state diagram in Fig. 1a; Fig. S1). Two-color fluorescence time-lapse imaging showed that the addition of poly(rU) RNA to PLP-RRP phase-separated mixture leads to the dissolution of PLP-RRP droplets and subsequent formation of RRP-RNA droplets (Fig. 2a, b, Movie S1). The dissolution of PLP-RRP droplets is preceded by a change in their color from yellow (RRP + PLP) to green (PLP), indicating that RRP is leaving these droplets and hence weakening the condensate network and leading to their dissolution (Fig. 2b). This condensate switching behavior (from PLP-RRP to RRP-RNA condensates) in the presence of RNA signifies a competition between RRP-PLP and RRP-RNA interactions, with RRP-RNA interactions being stronger than RRP-PLP interactions (Fig. 2a). To investigate whether this effect is specific to poly(rU) RNA, we repeated the same assay using poly(rA) RNA. Confocal video microscopy revealed that poly(rA) RNA is also able to induce the observed condensate switching by sequestering RRPs out of PLP condensates and subsequently forming RRP-RNA condensates (Fig. 2c). Collectively, these results suggest that RNA can induce a condensate-switching transition from PLP-RRP condensates to RRP-RNA condensates due to its superior interactions with the RRP.

**RNA triggers a de-mixing transition of RRP and PLP**. Our observations in Fig. 2 indicate that RNA can sequester RRP out of PLP condensates, which leads to the dissolution of PLP condensates. We, therefore, asked whether forming PLP-RRP

condensates at PLP concentration ($C_{PLP}$) greater than PLP saturation concentration ($C_{PLP} > C_{sat}$; the pink region in the state diagram in Fig. 1a; Fig. S1), would alter the observed RNA-triggered condensate switching effect (Fig. 2). Repeating our measurements under such conditions, we observed a de-mixing transition, where PLP-RRP condensates reorganized into homotypic PLP condensates and heterotypic RRP-RNA condensates in response to RNA addition (Fig. 3a, b, Movie S2, Fig. S16). The time-lapse images reveal that RNA sequesters RRP (red) from PLP-RRP droplets, reaffirming the apparent dominance of RRP-RNA interactions over RRP-PLP interactions. Following sample equilibration upon RNA addition, we observed that PLP condensates and the newly formed RRP-RNA condensates coexist in a multiphasic pattern where RRP-RNA droplets are distributed on the surfaces of PLP droplets (Fig. 3c). This multiphasic pattern was persistent throughout the sample, although a few small RRP-RNA droplets were present as isolated droplets without interacting with PLP droplets. To confirm that the multiphasic condensate formation is not an artifact due to the order of RNA addition, we mixed pre-formed PLP condensates ($C_{PLP} > C_{sat}$) with RRP-RNA condensates (prepared independently) and imaged them using a confocal microscope. We observed that PLP condensates coexist with RRP-RNA condensates in a multiphasic pattern that is reproducible irrespective of the method of sample preparation (Fig. S17) and is stable for more than 24 h (Fig. S18). We also confirmed that our results are not specific to poly(rU) RNA by repeating these experiments with a considerably shorter RNA chain rU40, poly(rA) RNA, and yeast total RNA (Fig. 3e, f, Fig. S19).

The coexisting condensates (e.g., PLP and RRP-RNA), although quite dynamic as indicated by their respective coalescence-induced condensate growth (Fig. S20), do not exchange their components. Based on these observations, we hypothesized that these coexisting condensates can support distinct microenvironments and differentially recruit biomolecules within[63,64] and that the RNA-induced condensate demixing may represent an active pathway for sorting those biomolecules in different condensates (Fig. 3g). Indeed, we observed that a short fluorescently labeled RNA molecule (5′-FAM-UGAAG-GAC-3′) and a disordered protein (Pol II CTD) that simultaneously partition into PLP-RRP droplets in absence of any RNA are differentially sorted into coexisting condensates in the presence of RNA. More specifically, we observed that the RNA molecules preferentially partition into the RRP-RNA droplet and the Pol II CTD is enriched in the PLP droplets (Fig. 3h). To determine whether RNA-induced demixing can be reversed, we tested the stability of demixed condensates in the presence of an RNA degrading enzyme, RNase-A. We observe that upon the addition of RNase-A, demixed PLP condensates and RRP-RNA condensates transition to well-mixed condensates (hosting both RRP and PLP) within three hours (Fig. 3i, j). Taken together, these results suggest that the competition between RRP-RNA and RRP-PLP intermolecular interactions in a ternary PLP-RRP-RNA mixture can give rise to a rich multiphasic behavior that constitutes condensate switching, condensate demixing, and biomolecule sorting (Figs. 2 and 3).

**Mixture composition tunes the morphology of coexisting PLP and RNA-RRP condensates**. Multi-phasic structures, which stem from the coexistence of multiple immiscible liquid phases[65–67], are hallmarks of several subcellular biomolecular condensates such as the nucleolus and stress granules[11,14,68]. For a four-component system (A, B, C, D) that undergoes phase separation into three phases- A (condensate), B + C (condensate), and D (dispersed liquid phase), the equilibrium morphology is

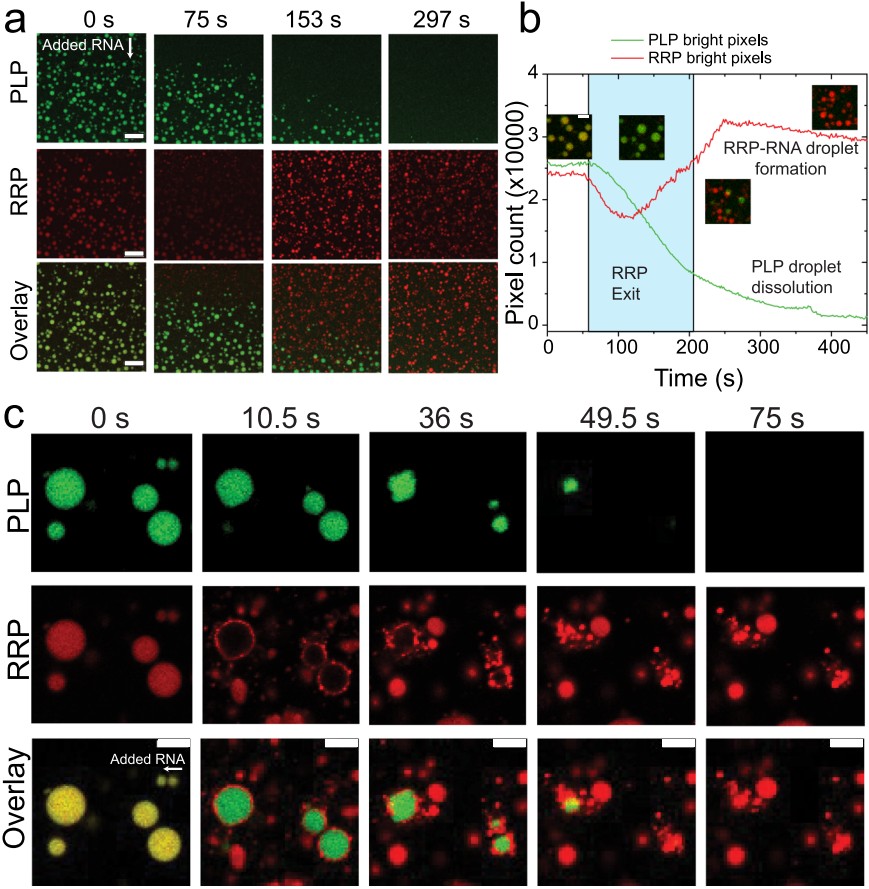

**Fig. 2 RNA induces condensate switching from PLP-RRP to RRP-RNA. a** Multicolor confocal fluorescence time-lapse images showing dissolution of PLP-RRP (RRP: FUS$^{RGG3}$) droplets and subsequent formation of RRP-RNA droplets upon addition of poly(rU) RNA. Scale bar = 20 μm. **b** A plot of the total area covered by condensates in the green (PLP) and red (RRP) channels for the data shown in (**a**) and in Movie S1. The areas were calculated by counting the green pixels (for PLP) and the red pixels (for RRP) and plotted as a function of time. The images indicate the various stages of sample evolution after RNA addition. The white region (left) indicates the time-window where PLP and RRP co-localize; the cyan shaded region indicates the time-window when RRP is leaving the PLP-RRP condensates; the white region (right) indicates the subsequent dissolution of PLP-RRP condensates and the formation of RRP-RNA condensates. Source data are provided as a Source Data file. Scale bar represents 5 μm. **c** Same assay as (**a**) but with poly(rA) RNA. (RRP: [RGRGG]$_5$). Scale bars represent 4 μm. All samples were prepared in 25 mM Tris-HCl, 150 mM NaCl, and 20 mM DTT buffer. PLP: FUS$^{PLP}$.

determined by the relative interfacial tensions ($\gamma_{A-D}$, $\gamma_{[B+C]-D}$, and $\gamma_{A-[B+C]}$) of the three liquid phases[11,15,16]. Based on the rank order of $\gamma_{A-D}$, $\gamma_{[B+C]-D}$, and $\gamma_{A-[B+C]}$, three configurations are possible (Fig. 4a): (i) condensates do not share any interface and remain separated ($\gamma_{A-[B+C]} > \gamma_{A-D} + \gamma_{[B+C]-D}$; non-engulfment); (ii) condensates partially merge in a way that they share a common interface but are still partly exposed to the dispersed liquid phase ($\gamma_{A-[B+C]} \sim \gamma_{A-D} \sim \gamma_{[B+C]-D}$; partial engulfment); and (iii) condensate B + C resides within the condensate A and remains completely separated from the dispersed liquid phase ($\gamma_{[B+C]-D} > \gamma_{A-D} + \gamma_{A-[B+C]}$ or vice-versa; complete engulfment).

Our experimental results in Fig. 1d–k showed that the surface architecture of RRP-RNA condensates is tuned by RNA-to-RRP stoichiometry, resulting in a switch-like change of PLP interactions with these condensates. This raises an interesting possibility of regulating interfacial interactions between RRP-RNA condensates and PLP condensates, given that RRP, but not RNA, multivalently interacts with PLPs. As such, increasing RNA concentration may lead to a controlled morphological variation in the coexisting PLP and RRP-RNA condensates. To test this possibility, we reconstituted the condensate pair by mixing homotypic PLP condensates with preformed RNA-RRP condensates prepared at a variable RNA-to-RRP stoichiometry. We chose RNA-to-RRP ratios such that they span the entire range of

the reentrant RNA-RRP LLPS regime (Fig. S21). We observed that the RRP-rich condensates ([RNA]:[RRP] = 0.1, 0.2, 0.75) are almost completely engulfed by the PLP homotypic droplets (Fig. 4b). On the contrary, RNA-rich condensates ([RNA]:[RRP] = 1.25, 2.5, and 5.0) are only partially engulfed by PLP droplets with a shared interface between the two condensate types that substantially decreases with increasing RNA concentration (Fig. 4b). To quantify the variation in the interfacial patterning of the two condensate types with changing RNA-to-RRP ratio, we estimated the contact angles for both PLP ($\theta_{PLP}$) and RRP-RNA ($\theta_{RRP}$) droplets using an image analysis approach (see Methods; Fig. 4c). The contact angles are the dihedral angles formed by the liquids at the three-phase boundary. Since the forces acting on the three-phase boundary should sum to zero (Neumann triangle), relations between the contact angles and the interfacial tensions can be derived[65,69,70]. Therefore, the cosine of a contact angle ($\theta$), can be expressed as a function of the interfacial tensions of the three liquid phases as[65,69]

$$\cos(\theta_{PLP}) = \frac{\gamma_{RRP+RNA}^2 - \gamma_{PLP}^2 - \gamma_{PLP-[RRP+RNA]}^2}{2\gamma_{PLP}\gamma_{PLP-[RRP+RNA]}} \quad (1)$$

A small $\theta_{PLP}$ indicates that the PLP phase is engulfing the RRP-RNA phase partially and the contact area between the two droplets is significant. On the other hand, a large value of $\theta_{PLP}$

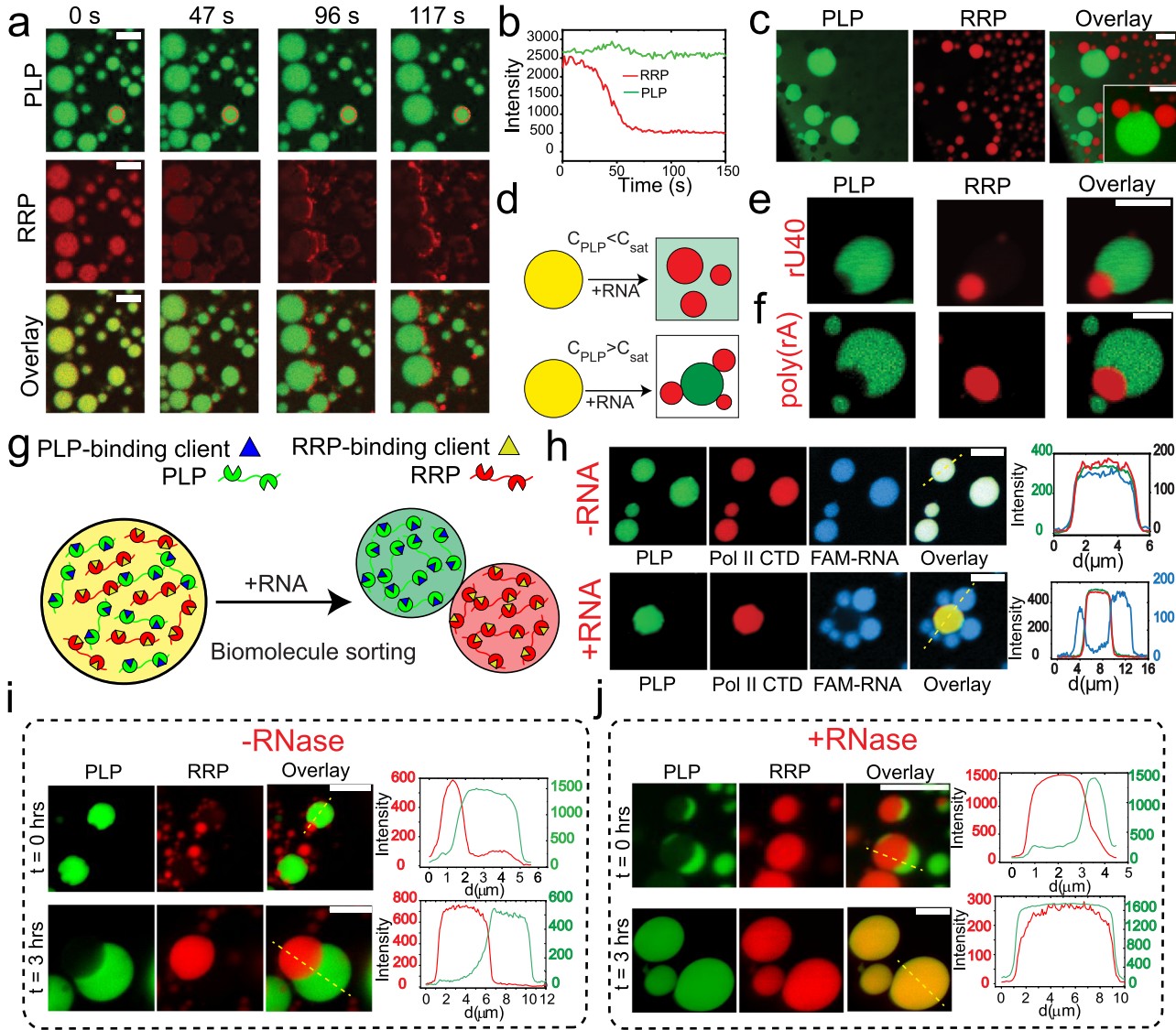

**Fig. 3 RNA causes the demixing of RRP and PLP. a** Fluorescence time-lapse images showing the sequestration of RRP (FUS^RGG3) from PLP-RRP droplets and the formation of RRP-RNA droplets upon addition of poly(rU) RNA. [PLP] = 250 μM; [FUS^RGG3] = 750 μM or 2.6 mg/ml; and poly(rU) RNA is added to a final concentration of 13 mg/ml. **b** PLP and RRP intensities as a function of time within a PLP-RRP condensate [red circle in (**a**)]. **c** Fluorescence images of coexisting PLP droplets and RRP-RNA droplets 20 min after RNA addition. [PLP] = 250 μM; [FUS^RGG3] = 1250 μM (4.3 mg/ml); and poly(rU) RNA is added to a final concentration of 10.8 mg/ml. **d** A schematic diagram summarizing the effect of RNA on PLP-RRP condensates. **e** Fluorescence microscopy images of coexisting PLP condensates (green) and RRP-RNA condensates (red), prepared using rU40 RNA. Each type of droplet was prepared independently at initial concentrations of [PLP] = 400 μM, [FUS^RGG3] = 4.0 mg/ml and [rU40] = 4.0 mg/ml and then mixed (1:1 vol/vol). **f** Fluorescence images of the coexisting PLP condensates (green) and RRP-RNA condensates (red) prepared using poly(rA) RNA. **g** A schematic showing that RNA-induced de-mixing of PLP-RRP condensate (yellow) into PLP (green) and RRP-RNA (red) condensates can sort diverse clients into different condensates. **h** Fluorescence micrographs and intensity profiles showing recruitment of a FAM-labeled short ssRNA and Alexa488-labeled Pol II CTD into PLP-RRP condensates in the absence of RNA (top). These RNA and polypeptide molecules are differentially sorted when poly(rU) RNA is added to the mixture (bottom)—FAM-RNA (blue) into RRP-RNA condensates and Pol II CTD (red) into PLP condensates. PLP condensates (PLP = 400 μM) were mixed (1:1 by volume) with a sample containing 4.0 mg/ml [RGRGG]_5 and 0.0 mg/ml (top) or 8.0 mg/ml (bottom) of poly(rU) RNA. **i, j** Fluorescence time-lapse images and intensity profiles (across the yellow dashed line) for coexisting PLP droplets and RRP-RNA ([RGRGG]_5-rU40) droplets in the absence (**i**) and presence of RNase-A (**j**). Both samples were prepared at [PLP] = 400 μM, [RGRGG]_5 = 1 mg/ml and [rU40] = 1 mg/ml. For the sample in (j), RNase-A concentration was 1.6 mg/ml. Buffer contains 25 mM Tris-HCl (pH = 7.5), 150 mM NaCl, and 20 mM DTT. PLP = FUS^PLP. Scale bar = 10 μm for (**a**, **c**) and 5 μm for (**e**–**j**).

indicates that the two phases do not have a significant preference for a shared interface (Fig. 4c, top panel). Remarkably, we observed a sigmoidal-like transition of PLP contact angle ($\theta_{PLP}$) as the RNA-to-RRP ratio is increased, while the RRP contact angle ($\theta_{RRP}$) remained unchanged (Fig. 4c, bottom panel). Thus, variation in $\theta_{PLP}$ quantifies the transition in the coexistence pattern observed in Fig. 4b with increasing RNA concentration.

According to Eq. 1, the observed variation in the PLP contact angle indicates a change in the relative rank order of the interfacial tensions in the three-phase system[65]. To confirm this finding, we performed computer simulations utilizing a fluid-interface modeling tool [Surface Evolver[71]]. Briefly, we created equal volumes of two immiscible liquids with specified interfacial tensions and minimized the total energy of the system by

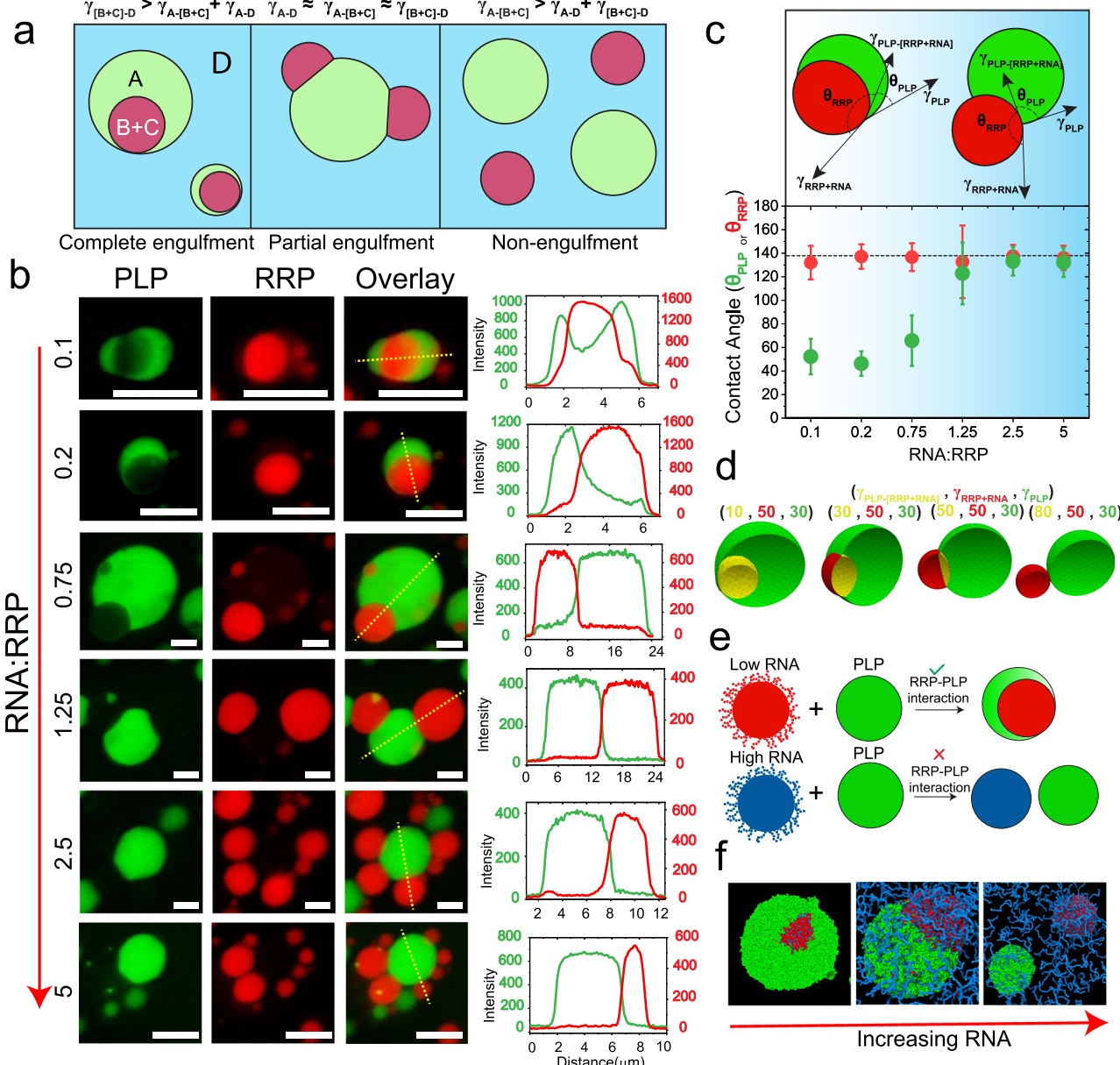

**Fig. 4 RNA-to-RRP ratio tunes the morphology of coexisting condensates. a** A schematic diagram showing that the relative ranking of interfacial tensions dictates the morphology of the biphasic condensates (A-droplet, B + C-droplet, D-dispersed phase). **b** Fluorescence microscopy images and intensity profiles for coexisting homotypic PLP droplets (green) and heterotypic RRP-RNA condensates (red) at different RNA-to-RRP ratios. Each type of droplet was separately prepared at initial concentrations of [PLP]=400 μM, [RGRGG]$_5$=4.0 mg/ml and variable poly(rU) RNA-to-RRP ratios (wt/wt), as indicated and then mixed (1:1 vol/vol). All samples were prepared in 25 mM Tris-HCl, 150 mM NaCl, and 20 mM DTT buffer. All Scale bars represent 5 μm. **c** Contact angle plot (bottom) for coexisting PLP ($\theta_{PLP}$ in green) and RRP-RNA ($\theta_{RRP}$ in red) condensates for all the samples shown in (**b**) ($n = 25$ droplets per sample). The dashed line represents the average value of $\theta_{RRP}$ across all samples. Data are presented as mean values ±1 s.d. (top) A schematic showing the coexisting condensates' morphology at low and high $\theta_{PLP}$. Color gradient (blue) represents the increasing RNA concentration. Source data are provided as a Source Data file. **d** Coexistence patterns of a model doublet-of-droplet as a function of interfacial tensions calculated using a fluid interfacial modeling tool (Surface Evolver[71]). **e** Proposed mechanism of RNA-mediated fluid-fluid interface regulation. At low RNA concentration, RNA-RRP condensates (red) are enriched with RRP chains on their surfaces, thus facilitating RRP-PLP interfacial binding and mediating a wetting behavior. At high RNA concentration, RRP-RNA condensate surfaces (blue) are enriched with RNA chains, limiting the available RRP molecules for PLP binding, which is responsible for minimal wetting behavior with PLP condensates (green). **f** Equilibrium MD snapshots at variable RNA-to-RRP mixing ratios (see Fig. S22b for the corresponding density profiles). (RRP = FUS$^{RGG3}$, red; RNA = poly(rU), blue; PLP=FUS$^{PLP}$, green). $C_{RRP}$ = 5.6 mg/ml, RNA-to-RRP ratio = 0.3, 1.8, 3.2 (respectively for the three snapshots shown), $C_{PLP}$ = 7.22 mg/ml.

modifying the shape of each liquid interface (see "Methods"; Fig. S22a). As the relative magnitude of the interfacial tension between the two condensates (i.e., $\gamma_{PLP-[RRP+RNA]}$) increased, we observed that the droplets transitioned from a completely engulfed morphology to a partially engulfed state and subsequently to a non-engulfing state (Fig. 4d), similar to our experimentally observed morphological transition with RNA concentration (Fig. 4b). By comparing the results in Fig. 4d with

those in Fig. 4b, we conclude that increasing the RNA concentration energetically destabilizes the shared interface between PLP droplets and RRP-RNA droplets.

A molecular mechanism that explains the effect of RNA on regulating the coexistence pattern can be deduced from the consideration that the surface organization of RNA-RRP condensates is altered as a function of the RNA-to-RRP mixing ratio (Fig. 1d, e). At low RNA-to-RRP ratios, the surface enrichment of free and/or partially condensed RRPs (Fig. 1d) confers a high propensity for the RRP-RNA condensates to favorably interact with the homotypic PLP condensates (Fig. 4e) through PLP-RRP interfacial binding (Fig. 1d–f). However, at high RNA-to-RRP concentration ratios, the surface of RRP-RNA condensates is enriched with free/partially condensed RNA chains (Fig. 1d), rendering the interfacial interactions between PLP and RRP-RNA droplets significantly less favorable (Fig. 4e; Figs. S5, S9). Consequently, the interfacial tension between the two types of condensates (PLP and RRP-RNA) is expected to increase. Therefore, as the RNA-to-RRP ratio increases, a progressive decrease in the contact area between the two condensates takes place (Fig. 4b–e). To test this idea, we used MD simulations utilizing homotypic PLP condensates and RRP (FUS$^{RGG3}$)-RNA condensates. The simulated equilibrium structures (Fig. 4f, Fig. S22b) suggest that with increasing RNA concentration, a progressive morphological transition occurs wherein the coexistence pattern of PLP and RRP-RNA droplets transitions from complete engulfment to partial engulfment to completely separated (non-engulfment) morphologies. We note that a similar morphological transition was recapitulated through Cahn-Hilliard fluid-interface simulations in a recent study, which suggested that differential interaction strengths between individual components in a ternary system can drive this process[64]. Taken together, these results further support that relative RNA concentration (i.e., RNA-to-RRP stoichiometry) plays a central role in determining morphologies of multiphasic condensates (Fig. 4b–d, f).

**Sequence-encoded protein–protein and protein–RNA interactions determine multiphasic condensate structuring.** Our results in Figs. 1 and 4 indicate that PLP-RRP interactions (or a lack thereof) at the liquid-liquid interface determines the stability of the interface between PLP condensates and RRP-RNA condensates. These results are consistent with the idea that the interfacial tension of a fluid-fluid interface is determined by the intermolecular interactions of the two given fluids at a known thermodynamic state[72]. Combining the results shown in Fig. 4 with those in Fig. 1, we propose that tuning the molecular interactions between components present on the surfaces of coexisting droplets is sufficient to control the multiphasic coexistence pattern (Fig. 4e). To experimentally test this idea, we designed an all K-variant of RRP (KRP: [KGKGG]$_5$, Table S2) that selectively weakens the PLP-RRP interactions while preserving the ability to phase separate with RNA. The R-to-K substitution is designed based on prior studies showing that (i) lysine residues have a much lesser potency (as compared to arginine residues) to interact with tyrosine and other π-rich amino acids/nucleobases[21,26,28,29,73–75], and (ii) KRPs can phase separate with RNA via ionic interactions since lysine is expected to carry a similar charge to that of arginine[28]. Indeed, our state diagram analysis indicates that KRP has no impact on PLP phase separation, even at KRP concentrations that are 20 times more than the PLP concentration (Fig. 5a). Simultaneously, we confirmed that the KRP phase separates with RNA under similar conditions as the RRP (Fig. S21)[28]. We further verified the lack of PLP-KRP interactions by confocal microscopy experiments,

which reveal that both PLP partition coefficient and PLP apparent diffusion rate within PLP condensates remain unchanged with increasing KRP concentrations in solution (Fig. 5b, Fig. S23; compare these results with Fig. 1a–c). Consistent with these results, PLP partitioning within KRP-RNA condensates remained very low (<1.0) and unchanged at variable RNA-to-KRP mixing ratio (Fig. S24), further confirming the absence/insignificance of PLP-KRP (and PLP-RNA) interactions. Therefore, R-to-K substitutions successfully abrogate PLP-RRP interactions by eliminating Arg-Tyr interactions[21].

Next, we probed for the coexistence pattern of PLP and KRP-RNA condensates. As expected, we observed that the two types of droplets (PLP homotypic and KRP-RNA heterotypic) do not share any interface (non-engulfment) at all the tested RNA-to-KRP mixing ratios (Fig. 5c, Fig. S21). These results confirm that the stability of a shared interface is critically contingent on the presence of PLP-RRP interactions at the surfaces of these two condensates. Consistent with this, we further observed that a π-rich peptide variant of the KRP, [KGYGG]$_5$, which enhances the PLP phase separation (Fig. S25a), restores the partial engulfing morphology of ternary PLP-KRP-RNA condensates (Fig. S25b). Overall, these findings suggest that sequence-encoded molecular interactions at the liquid–liquid interface between two condensed phases have a direct role in dictating the respective morphological pattern of these coexisting phases (Fig. 5a–d).

According to our model and experimental data (Fig. 4), RNA-rich RRP condensates do not share a significant interface with the PLP condensates due to the absence of PLP-RNA interactions (Fig. 4e). We posit that the addition of an RNA-binding module to the PLP could aid in lowering the interfacial tension between the condensates and thereby forming a shared interface with the RNA-rich RRP condensates. We tested this idea by utilizing the full-length FUS (Fused in Sarcoma), which is composed of a PLP (FUS$^{PLP}$) and RNA-binding Arg-rich domains (FUS$^{RBD}$) (Fig. 5e). The full-length FUS (FUS$^{FL}$) showed enhanced partition into RRP-RNA condensates (as compared to FUS$^{PLP}$, Fig. 1h) across all RNA-to-RRP mixing ratios (Fig. S26). In addition, MD simulations indicate that FUS$^{FL}$ is recruited on the surface of RRP-RNA droplets at both low and high RNA concentrations (Fig. S27). These results confirm that FUS$^{FL}$ interacts with both RRP-rich and RNA-rich condensates. Consistent with our prediction, confocal microscopy imaging of coexisting homotypic FUS$^{FL}$ droplets and heterotypic RRP-RNA droplets revealed that FUS$^{FL}$ droplets completely engulf the RNA-RRP droplets (i.e., a significant interfacial contact area) across all tested RNA-to-RRP mixing ratios (Fig. 5f and Fig. S28). Furthermore, weakening the FUS-RRP interactions by replacing RRP with its R-to-K variant (KRP: [KGKGG]$_5$) resulted in a partially engulfed morphology (Fig. 5g and Fig. S29). Interestingly, these FUS-KRP-RNA multiphasic condensates displayed morphologies that are reminiscent of Janus spheres[76,77] with two compositionally distinct lobes (Fig. 5g and Fig. S29). Taken together, these results confirm that modular intermolecular interactions directly govern the coexistence pattern for multiphasic condensates by controlling the dominant interactions at the liquid–liquid interface.

**Stability diagram of multiphasic condensates establishes a link between intermolecular interactions, interfacial tensions, and experimentally observed condensate morphologies.** Our experimental results and computational modeling presented here collectively suggest a clear relation between the microscopic intermolecular interactions and the mesoscopic multiphase structuring that transcends length-scales. By comparing our experimental and MD simulation data with our fluid-interface modeling results (Fig. 4), we infer that the interplay between

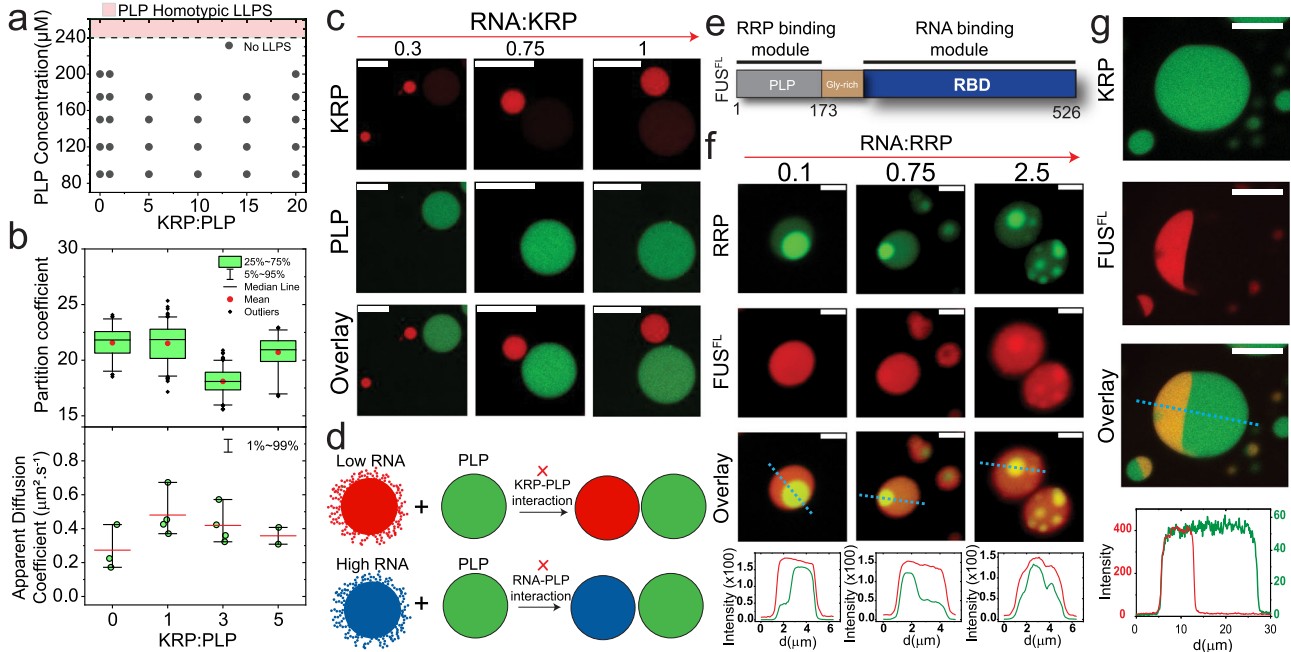

**Fig. 5 Intermolecular interactions between RNA and protein components tune the morphology of coexisting condensates. a** State diagram of PLP-KRP mixtures as a function of KRP-to-PLP ratio (mole: mole), showing that KRP, [KGKGG]$_5$, does not affect PLP phase-separation (compare with Fig. 1a for RRP-PLP mixtures). **b** PLP partition coefficient ($n = 60$ droplets per sample) and diffusion rate ($n = 3$ droplets per sample) in PLP-KRP condensates as a function of KRP-to-PLP mixing ratio (mole: mole). The error bars represent the range of data (1–99%) for the bottom panel while the red line represents the mean value. Source data are provided as a Source Data file. **c** Fluorescence images showing that the morphology of coexisting PLP homotypic condensates and KRP-RNA condensates is non-engulfing and does not vary with RNA-to-KRP stoichiometry. Each type of droplet was separately prepared at initial concentrations of [FUS$^{PLP}$] = 400 μM, [KGKGG]$_5$ = 4 mg/ml, and variable poly(rU) RNA-to-KRP ratios (wt/wt), as indicated, and mixed (1:1 vol/vol). **d** A schematic diagram showing that due to insignificant KRP-PLP interfacial interactions, the PLP homotypic and KRP-RNA heterotypic condensates do not share any interface (non-engulfment) at both low and high RNA. **e** Domain architecture of FUS$^{FL}$ showing both PLP and RBD modules. **f** Fluorescence microscopy images and intensity profiles for coexisting homotypic FUS$^{FL}$ droplets (red) and heterotypic RRP-RNA condensates at different RNA-to-RRP ratio. Each type of droplet was separately prepared at initial concentrations of [FUS$^{FL}$] = 21.3 μM, [FUS$^{RGG3}$] = 1 mg/ml and variable poly(rU) RNA-to-RRP ratios (wt/wt), as indicated, and mixed (1:1 vol/vol). **g** Fluorescence micrographs and intensity profiles for Janus droplets formed by coexisting homotypic FUS$^{FL}$ droplets (red) and heterotypic KRP-RNA condensates (green). Each type of droplet was separately prepared at initial concentrations of [FUS$^{FL}$] = 22 μM, [KGKGG$_5$] = 4 mg/ml and poly(rU) = 3 mg/ml and mixed (1:1 vol/vol). All samples were made in a buffer containing 25 mM Tris-HCl, 150 mM NaCl, and 20 mM DTT. Scale bars represent 10 μm for (**c**, **g**) and 2 μm for (**f**).

various intermolecular interactions amongst components (PLP, RRP, and RNA) determines the relative rank order of interfacial tensions between the coexisting liquids. To verify this idea, we consider the equilibrium configurations of two immiscible droplets (PLP droplet and RRP-RNA droplet) in water as a function of the relative values of their interfacial tensions. Since all of our multiphasic condensate analysis was performed at the same conditions for PLP condensates, we chose to fix the interfacial tension of PLP droplets ($\gamma_{PLP}$) and vary the interfacial tension of RRP-RNA droplets ($\gamma_{RRP+RNA}$) and the interfacial tension between PLP droplets and RRP-RNA droplets ($\gamma_{[RRP+RNA]-PLP}$). Employing fluid-interface modeling, we construct a stability diagram (Fig. 6a) that marks the boundaries between three distinct morphological states (non-engulfing, partially engulfing, and complete engulfing) in terms of the relative values of these interfacial tensions (i.e., in terms of $\gamma_{RRP+RNA}/\gamma_{PLP}$ and $\gamma_{[RRP+RNA]-PLP}/\gamma_{PLP}$). This stability diagram identifies possible transition pathways between different ternary condensate morphologies, which were observed in our experiments (Fig. 6a). We note that our stability diagram is able to capture all the variable morphologies observed in the experiments, indicating that tuning interfacial tensions may be sufficient to encode diverse multiphasic patterning of the two-condensate system.

Subsequently, based on our experimental data and MD simulation results (Figs. 4 and 5), we propose a mechanistic model that connects the RNA-dependent RRP and PLP interactions with the fluid-fluid interfacial interactions in the mesoscale. The transition between total engulfment to partial and non-engulfment with increasing RNA concentrations (Fig. 4b) can be recapitulated in simulations by varying $\gamma_{[RRP+RNA]-PLP}/\gamma_{PLP}$ while keeping the other interfacial tensions unchanged (Fig. 6b; Pathway-A). By casting this transition (Pathway-A) in Fig. 6b onto the experimental data shown in Fig. 4b, we can deduce that increasing the RNA-to-RRP ratio increases the interfacial energy between RRP-RNA droplets and PLP droplets. This is expected since RNA-PLP interactions (which dominate the interface at excess RNA conditions) are significantly weaker than RRP-PLP interactions (which dominate the interface at excess RRP conditions, Fig. 1a–c and Fig. S5). However, we note that Pathway-A may not be unique and a more complex pathway (such as Pathway-B) is also possible due to a simultaneous change in $\gamma_{RRP+RNA}$, and hence $\gamma_{[RRP+RNA]}/\gamma_{PLP}$, as a function of RNA. Replacing arginine with lysine abolishes RRP-PLP interactions, leading to an overall non-engulfment morphology (Fig. 6a; Pathway-B, Fig. 5c). Next, the covalent coupling of PLP with an RNA-binding module creates a bi-valent scaffold (i.e., FUS$^{FL}$) with independent sites for RRP and

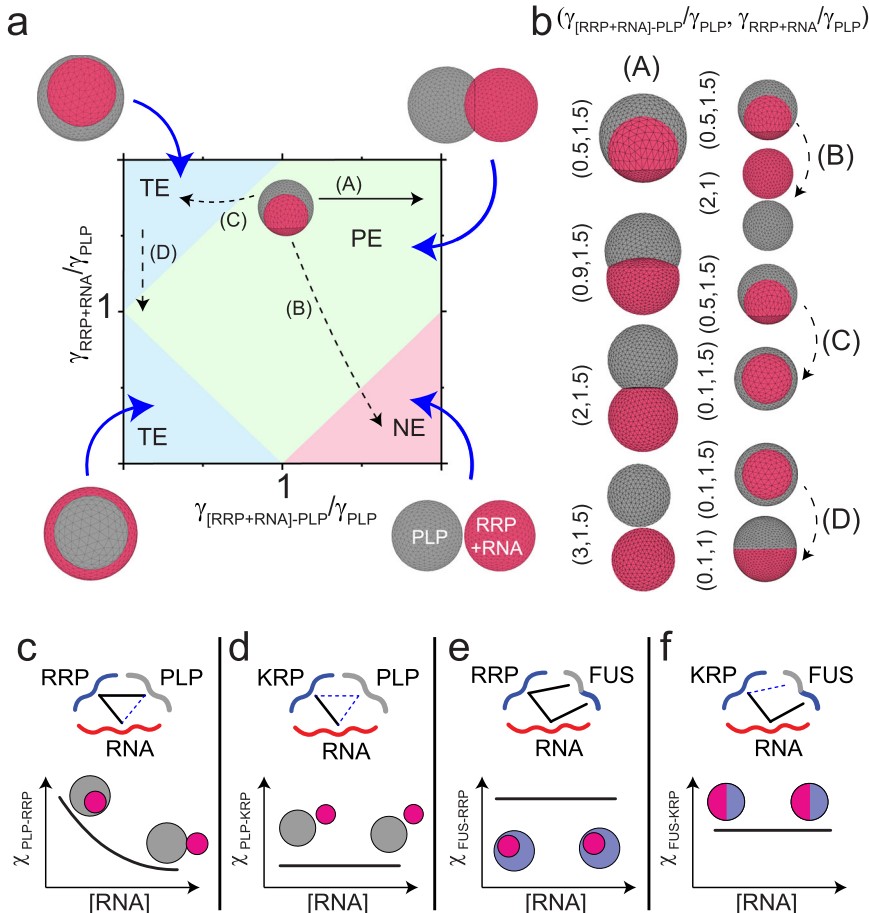

**Fig. 6 The stability diagram of a pair of coexisting condensates provides a link between intermolecular interactions and multiphasic morphology.**
**a** Stability diagram for PLP droplets coexistence pattern with RRP-RNA droplets from fluid-interface modeling simulations. In these simulations, $\gamma_{PLP}$ was fixed and $\gamma_{RRP+RNA}$ and $\gamma_{[RRP+RNA]-PLP}$ were varied. The shaded regions mark the different morphological states: total engulfment (TE), partial engulfment (PE), and non-engulfment (NE). The solid black arrow indicates a continuous morphological transition with RNA dosage as described in the text. The dashed lines correspond to discrete transitions due to sequence variations. **b** Simulation strips showing the continuous and discrete morphological transitions as the values of the interfacial tensions are varied along with the corresponding arrows in the stability diagram in (**a**). **c–f** Schematic diagrams showing the interactions between the ternary components as well as the observed morphology as a function of RNA concentration. The schematic plots show how the interactions between the two types of droplets ($\chi$) are expected to change as a function of RNA dosage. The solid lines in the schematic interaction diagrams indicate strong interactions while the dashed lines indicate weak and/or absent interactions.

RNA binding, which manifests in completely engulfed ternary morphologies in an RNA-independent manner (Fig. 6a-Pathway-C; Fig. 5e, f). Subsequent weakening of RRP-FUS interactions via R-to-K substitutions leads to the formation of partially engulfed Janus-like morphologies (Fig. 6a-Pathway-D, Fig. 5g and S29). Taken together, these results enable us to reliably correlate sequence-encoded intermolecular interactions with multiphasic behavior, leading to coexistence patterns that can be controlled by sequence perturbations as well as mixture composition (Fig. 6c–f).

## Discussion
Intracellular biomolecular condensates encompass a plethora of multivalent proteins and RNAs, which constitute a dense intermolecular interaction network[78]. The fluid-structure of these multi-component condensates typically shows coexisting layers of liquid phases as opposed to a well-mixed isotropic liquid condensate. Here, we report experimental and computational evidence of a minimal biomolecular condensate forming ternary system displaying a rich variety of multiphasic structuring and spatial organization that is primarily governed by composition-

dependent and sequence-encoded intermolecular interactions. First, we show that varying RNA-to-RRP mixing ratio changes the mesoscale organization of the condensed phase at the single condensate level (Fig. 1). The distinction between the core and surface compositions of RNA-RRP condensates can be attributed to differential solvation of non-stoichiometric RRP-RNA complexes at compositionally disproportionate mixtures[53] (Fig. 1d). For example, in RNA-rich condensates, an unbound or partially complexed RNA chain is expected to have a larger effective solvation volume as compared to fully complexed RRP-bound RNA chains, resulting in free RNA chains being preferentially positioned on the condensate surface[53,79] (Fig. 1d, e). A similar argument can also be made for RRP-rich condensates, leading to a composition-dependent tuning of the condensate surface architecture. Such organizational tuning is perceived to be a general phenomenon for heterotypic condensates and is likely to be functionally important in regulating client recruitment and controlling their spatial sub-organelle localization (Fig. 1e–k).

Second, we demonstrate that multiphasic condensates can form in a minimal ternary system via molecular competition for a shared binding partner. In the case of the PLP-RRP-RNA system,

RRP-RNA interactions dominate over PLP-RRP and this competition leads to an RNA-induced demixing of PLP and RRP into two immiscible phases from a single associative well-mixed RRP-PLP condensed phase (Fig. 3). The coexisting droplets offer distinct microenvironments and display selective client partitioning. We envision that the sequence and structure of RNA would strongly regulate the extent of this demixing phenomenon. We also speculate that competitive inhibition of aberrant intermolecular interactions between protein and RNA components can provide an attractive route to target certain biomolecular condensate microenvironments in human pathologies.

Third, we show that the coexistence patterning between homotypic PLP condensates and RNA-RRP condensates is directly related to the PLP-RRP interactions at the fluid-fluid interface (Figs. 4, 5). More specifically, our experiments and simulations suggest that PLP-RRP interactions govern the thermodynamic stability of a shared interface between PLP condensates and RRP-RNA condensates. As such, perturbation of PLP-RRP interactions on the condensate surface affects the inter-condensate interactions and hence, the interfacial tensions of the coexisting liquid phases. We report two mutually exclusive mechanisms to control the stability of a shared interface: (1) RNA dose-dependent regulation of PLP-RRP interactions through molecular competition (Fig. 4), and (2) perturbation of protein–protein intermolecular interaction network via RRP and PLP sequence variations, which in turn eliminates or enhances the interfacial interactions between PLP condensates and RNA-RRP condensates (Fig. 5). We note that our proposed mechanism 1 is unique to heterotypic condensates where the surface composition of the condensate can be distinct from the condensate core (Fig. 1). We further note a distinction between the composition-dependent regulation of interfacial energies in our ternary system, which is exclusively comprised of intrinsically disordered polymers, and a recently suggested mechanism for the coexistence of stress granules (SGs) and P-bodies (PBs)[31], where a distinct protein with a common preference for both SGs and PBs acts as a bridge between the two condensate types and controls the shared interfacial area. In the latter case[31], the relative amount of the bridging protein is an important variable in dictating the multiphase coexistence pattern. However, our results presented here for RNA-RRP condensates' multiphasic patterning with PLP condensates reveal a unique role of the condensates' surface organization, which can be manipulated by varying the mixture composition. This is likely to be a direct result of the existence of a structural continuum[80] in the ensemble of RRP-RNA complexes where the stoichiometry of the resulting complexes is sensitively dependent on the mixture composition[28,53,81,82]. In such a case, an RRP-rich protein–RNA complex, but not an RNA-rich protein–RNA complex, can act as an emergent molecular bridge between RRP-RNA condensates and PLP condensates (Fig. 4e). This allows control over the coexistence patterns in our minimal system without additional bridging proteins. Lastly, LLPS-driving proteins often feature modular architecture with both low-complexity disordered domains and structured modules. The presence of these structured domains (such as RNA recognition motifs) is expected to alter the multiphasic behavior and condensate properties, especially when these domains are involved in the interactions stabilizing the condensates[11,83].

Together, our presented experimental and computational results suggest that competing protein–protein and protein–RNA interactions are a regulatory paradigm for the organization of multiphasic biomolecular condensates. They also provide simple physical rules to utilize the phase separation of ternary biopolymer mixtures to create soft Janus-like particles with tunable morphologies in a stimuli-responsive fashion.

## Methods

**Protein expression and purification.** A list of the proteins used in this study is shown in Table S1. Codon optimized proteins of interest were gene-synthesized by GenScript USA Inc. (Piscataway, NJ, USA). The plasmid vector was a gift from Scott Gradia (Addgene plasmid # 29706). Proteins were expressed and purified using affinity chromatography as described in our earlier work with one modification[84]. Cells were lysed using a sonicator for 2 min (Branson Digital Sonifier 450, 3 mm tapered microtip, 50% amplitude, 10 s ON/ 50 s OFF) in an ice bath.

**Fluorescence labeling.** The cysteine-containing variants of the proteins were gene synthesized by GenScript USA Inc. (Piscataway, NJ, USA) through site-directed mutagenesis. The proteins were expressed and purified using an identical protocol as described above with one modification: all buffers contained 1 mM DTT to prevent cysteine cross-linking. The protein samples were fluorescently labeled with either Alexa488 dye, Alexa594 dye, or Cy5 dye (C5-maleimide derivative, Molecular Probes) as described in the manufacturer protocol. The His6-MBP-N10 tag was removed by the action of TEV protease (TEV: protein = 1:25 v/v) for 1 h at 30 °C. Ni-NTA beads (ThermoFisher Scientific; Cat# 88223) were used to separate the tag from the proteins. The cleaved proteins were diluted in 25 mM Tris HCl (pH 7.5), 125 mM NaCl (final concentration: 2-10 μM), and stored as aliquots at −80 °C. The labeling efficiency for all samples was observed to be ≥65% (UV–Vis absorption measurements). All peptides [purchased through GenScript USA Inc. (Piscataway, NJ, USA)] contained a C-terminal cysteine which was used for site-specific labeling with Alexa488 or Alexa594 dyes using the same protocol as described in our earlier work[28,49,85,86]. A list of the proteins used in this study is shown in Table S1.

**Peptide and RNA stock preparation.** All the peptides used ([RGRGG]₅, [KGKGG]₅, [KGYGG]₅, RGG-3 domain of FUS) were purchased from Genscript USA Inc. (NJ, USA). All peptides contained a C-terminus cysteine for site-specific peptide labeling. Peptide stock solutions were prepared in RNase-free water (Santa Cruz Biotechnology) with 50 mM dithiothreitol (DTT). Polyuridylic acid [poly (rU); Sigma-Aldrich; molecular weight = 600–1000 kDa], Polyadenylic acid [poly (rA); Sigma-Aldrich; molecular weight = 100–500 kDa], Poly(phosphate) [poly(P) p100, medium-chain; Kerafast Inc; molecular weight = 11.5 kDa], custom-synthesized RNA oligomer poly(rU)-rU40 [40 nucleobases; Integrated DNA Technologies (IDT); molecular weight = 12185 Da], yeast total RNA (Sigma-Aldrich)] and FAM-labeled RNA oligomer-rU10 [Integrated DNA Technologies (IDT)] were reconstituted in RNase-free water. The concentration of all RNAs was calculated from their respective measured absorbance at 260 nm in a UV–Vis spectrophotometer (Nanodrop oneC). Both RNA and peptide stock solutions were stored at −20 °C. Before sample preparation for experiments, the RNA [poly(rU), poly(rA), and rU40] stock solutions were checked for any aggregates using bright-field microscopy. Nucleic acid staining dye SYTO13 was purchased from Thermo Fisher Scientific Inc.

**State diagram analyses.** State diagrams for all FUS^PLP-peptide mixtures were determined using optical microscopy. Before sample preparation, FUS^PLP was buffer exchanged (to remove the glycerol used in the storage buffer) into 25 mM Tris-HCl buffer (pH 7.5) at room temperature. This is followed by the removal of His6-MBP-N10 tag using TEV protease (TEV:protein = 1:25 v/v) in a 25 mM Tris HCl (pH 7.5) buffer containing 150 mM NaCl for 1 h at 30 °C. Samples for phase diagram analyses were prepared at room temperature at the desired FUS^PLP and peptide concentrations in a 25 mM Tris-HCl buffer containing 150 mM NaCl and 20 mM DTT (pH 7.5). Samples were then placed onto a Tween20-coated (20% vol/vol) microscope glass slide and loaded under a Zeiss Primovert inverted iLED microscope (×40 or ×100 objective). Images were captured using a Zeiss Axiocam 503 monochrome camera. Samples were kept covered with a glass cover to prevent concentration fluctuations due to evaporation and monitored for 2–5 min for droplet formation. The mixture was marked as LLPS or no LLPS depending on the clear existence of visible droplets throughout the microscopic field of view. The state diagram for FUS^PLP-poly(rU) mixtures was also obtained similarly except the sample buffer used was 25 mM Tris HCl (pH 7.5) buffer, 150 mM NaCl without any DTT. The samples for the RRP-RNA state diagram were also prepared at room temperature in 25 mM Tris-HCl buffer containing 150 mM NaCl and 20 mM DTT (pH 7.5) and were analyzed for the existence of droplets similarly as stated above.

**Apparent diffusion coefficient measurement using FRAP.** Zeiss LSM710 laser scanning confocal microscope with a ×63 oil-immersion objective (Plan-Apochromat 63x/1.4 oil DIC M27) was used for fluorescence recovery after photobleaching (FRAP) experiments. Phase-separated FUS^PLP-peptide samples were prepared at a fixed FUS^PLP concentration (280 μM) and varying peptide concentration in a buffer containing 25 mM Tris-HCl,150 mM NaCl, and 20 mM DTT (pH 7.5). These samples correspond to the FUS^PLP concentrations above the homotypic phase-separation threshold for FUS^PLP as shown in the FUS^PLP-peptide state diagrams (Figs. 1a and 5a in the main-text). Sample preparation of FUS^PLP-peptide mixtures for FRAP was done similarly as described above for the state diagram analyses except for the addition of fluorescent probes. Approximately 1% (labeled-to-unlabeled ratio) of Alexa488-labeled FUS^PLP (excitation/emission

wavelengths; 488/503–549 nm) and Alexa594-labeled peptides (excitation/emission wavelengths; 595/602–632 nm) were used within the unlabeled protein-peptide mixtures. Samples were then placed inside a Tween20-coated (20% vol/vol) Nunc Lab-Tek Chambered Coverglass (ThermoFisher Scientific Inc.) for imaging and FRAP assays. For FRAP experiments, a circular region of interest was bleached with 100% power for ~2–6 s which was followed by an imaging scan for 300 s. The recorded Alexa488-labeled FUS$^{PLP}$ intensity values from the bleached ROI were then corrected for photofading, normalized, and fitted with a 2D diffusion model to obtain the recovery half time $\tau_{1/2}$ as described in our earlier work[84]. The apparent diffusion coefficient ($D$) was calculated using the formula[87]

$$D = \frac{R^2}{4\tau_{1/2}} \qquad (2)$$

where $R$ is the radius of the bleaching ROI (Figs. S3 and S23). The apparent diffusion coefficient was averaged for several samples (see "Statistics and reproducibility" section). Interval dot plots were plotted for comparison of apparent diffusion coefficients of FUS$^{PLP}$ within FUS$^{PLP}$-peptide condensates at different peptide concentrations.

**Partition coefficient measurements**. Images for partition analysis were collected using the same instrument as for the FRAP measurements above. The same samples were used for FUS$^{PLP}$-peptide mixtures as described above in the FRAP section. All the confocal images were collected within 30 min of sample preparation. The partition coefficient ($k$) was calculated by dividing the mean intensity of Alexa488-labeled FUS$^{PLP}$ per unit area inside the droplet by the mean intensity per unit area in the external dilute phase $\left(k = \frac{I_{in}}{I_{out}}\right)$. Images for the partition of FAM-labeled RNA oligomer-rU10 into FUS$^{PLP}$ condensates in the presence and absence of poly(rU) RNA were collected using a laser scanning confocal microscope (LUMICKS$^{TM}$ C-trap, 60x water-immersion objective). Samples were prepared at concentrations mentioned in the relevant figure legends (Fig. S6).

**Client recruitment assay**. Phase-separated samples were prepared at a fixed concentration (1.0 mg/ml) of the peptide (FUS$^{RGG3}$ or [KGKGG]$_5$) and variable concentrations of poly(rU) RNA as mentioned in the text or the figure legends. The concentrations of poly(rU) RNA were chosen such that the RNA-to-peptide ratio maps the left, right, and peak points on the turbidity plots of respective peptide and RNA (Fig. S21). To measure the recruitment of different clients in peptide-poly(rU) droplets, ~500 nM of Alexa-488 labeled clients (FUS$^{PLP}$; EWSR1$^{PLP}$; Pol II CTD; BRG1$^{LCD}$; FUS$^{FL}$) were added to the sample mixture (Fig. 1g–k, main-text). The sample also contained 1% (labeled: unlabeled ratio) of Alexa594-labeled peptides for visualization of the condensates. The samples were prepared in 25 mM Tris-HCL (pH 7.5), 150 mM NaCl, and 20 mM DTT buffer. The order of addition of different components during sample preparation was buffer, peptide, client, and poly(rU). The sample was placed inside a Tween20-coated (20% vol/vol) 8-well Nunc Lab-Tek chambered coverglass and images were collected using Zeiss LSM710 laser scanning confocal microscope with ×63 oil-immersion objective (Plan-Apochromat 63×/1.4 oil DIC M27). The recruitment assays for BRG1$^{LCD}$ and FUS$^{FL}$ were collected using LUMICKS$^{TM}$ C-trap, ×60 water-immersion objective wherein the sample was inserted in a Tween20-coated (20% vol/vol) 25 mm × 75 mm × 0.1 mm custom-made flow chamber. All the images were collected within 1 h of sample preparation. The client recruitment was quantified using the client partition coefficient ($k$) within the peptide-RNA droplets. The partition coefficient ($k$) was calculated by dividing the mean intensity of Alexa 488-labeled client per unit area inside the droplet by the mean intensity per unit area in the external dilute phase $\left(k = \frac{I_{in}}{I_{out}}\right)$. FUS$^{PLP}$ recruitment into FUS$^{RGG3}$ and poly(P) droplets were performed similarly as described above for FUS$^{PLP}$ recruitment in FUS$^{RGG3}$- poly(rU) droplets.

**RNA-mediated PLP-RRP condensate switching and demixing assays**. FUS$^{PLP}$-RRP (RRP = FUS$^{RGG3}$ or [RGRGG]$_5$) mixtures were prepared as described in the state diagram and the FRAP experiment sections. FUS$^{PLP}$ and RRP concentrations were chosen within the green/pink region in the state diagram (Fig. 1a main-text; and Fig. S1) above and below the saturation concentration of FUS$^{PLP}$ homotypic phase-separation in respective samples as described in the text. Samples were prepared in a buffer containing 25 mM Tris-HCl (pH 7.5), 150 mM NaCl, and 20 mM DTT and ~1% (labeled: unlabeled ratio) of Alexa 488-labeled FUS$^{PLP}$ (Figs. 2 and 3a–c, main-text) and Alexa 594-labeled peptide were added for fluorescence microscopy. Each sample was placed inside the Tween20-coated Nunc Lab-Tek chambered coverglass and loaded onto a Zeiss LSM710 laser scanning confocal microscope. The objective was focused on a suitable position in the middle of the sample with droplets. The Zeiss software was set to acquire time-lapse images continuously every 1.6 s before RNA addition. Once imaging is started, a 0.7–1 µl drop of RNA [poly(rU) or poly(rA)] stock solution was added to the sample using a pipette far from the image acquisition spot to a final concentration of 2.5 or 5 times (as mentioned in the appropriate figure legends) that of RRP concentration in the sample (wt/wt). Time-lapse images were acquired until the droplets equilibrated after RNA addition. A control experiment with an identical volume of buffer addition instead of RNA addition was performed to ascertain that

the changes seen in the FUS$^{PLP}$-RRP droplets were not due to concentration fluctuations (Fig. S16). Time-lapse images for the control experiment were captured using a Zeiss Axiocam 503 monochrome camera mounted on a Zeiss Primovert inverted iLED microscope (×40 objective).

**Preparation and imaging of multiphasic condensates**. Before sample preparation, all the proteins were buffer exchanged to remove the glycerol present in the storage buffer. FUS$^{PLP}$ was buffer exchanged in the same way as mentioned in the state-diagram analyses section while full-length FUS was buffer exchanged into a buffer constituting 25 mM Tris-HCl (pH 7.5) and 150 mM NaCl (pH 7.5). Next, the His6-MBP-N10 tag was cleaved using TEV protease (1:25 volume ratio-TEV: protein) for 1 h at 30 °C. Homotypic FUS$^{PLP}$/ FUS$^{FL}$ droplets were formed at room temperature at concentrations (FUS$^{PLP}$ = 400–500 µM; or FUS$^{FL}$ = 21 µM) well above their respective homotypic phase-separation thresholds[84]. 1–2% of the labeled protein was added to the sample of unlabeled proteins. For the fluorescent labels, we used Alexa488-labeled (Figs. 3e, f, 4b, and 5c, g, main-text) or Cy5-labeled (Figs. 3h and 5f, main-text) FUS$^{PLP}$ as well as Alexa488-labeled FUS$^{FL}$. In some instances, FUS$^{PLP}$ was used to visualize FUS$^{FL}$ condensates (Fig. 5f, g, main-text). The fluorescent probes for the supplementary figures are indicated in the appropriate figure legends. In parallel to this, peptide-RNA droplets were prepared in a separate tube with a fixed concentration of the peptide (as mentioned in respective figure legends) and variable concentration of poly(rU) RNA. The concentrations of poly(rU) RNA were chosen such that poly(rU)-to-peptide ratios map the left, right, and peak points on the turbidity plots of respective peptide and poly(rU) mixtures (Fig. S21). Approximately 500 nM of Alexa594-labeled peptides were used for fluorescence imaging. The buffer used for the samples contained 25 mM Tris-HCl, 150 mM NaCl, and 20 mM DTT (pH 7.5). These preformed peptide-RNA droplet samples were then mixed 1:1 (v/v) with homotypic FUS$^{PLP}$/FUS$^{FL}$ droplet samples. The resulting mixture containing the two types of droplets was then placed at the center of a Tween20-coated (20% v/v) 25 mm × 75 mm × 1 mm glass slide. The sample was then sealed with an 18-mm square coverslip of 0.1-mm thickness using double-sided tape. The resulting protein and peptide-RNA droplet samples were imaged using a laser scanning confocal microscope (LUMICKS$^{TM}$ C-trap, ×60 water-immersion objective). All the images were collected within one hour of sample preparation. The same method of preparation and imaging was used for rU40 RNA-peptide multiphasic condensates (Fig. 3e). Multiphasic condensates using yeast total RNA (Fig. S19) were prepared by mixing all components into a test tube and imaged using the same instrument as above.

**Contact angle analysis**. Contact angles between the co-existing PLP (FUS$^{PLP}$) droplets and RRP-RNA ([RGRGG]$_5$-poly(rU) RNA) droplets for the various samples were measured manually using the Angle Tool in Fiji-ImageJ[88]. Three tangent lines were drawn; (a) a tangent at the PLP-solvent interface, (b) a tangent at the interface between the RRP-RNA droplet and the solvent, and (c) a tangent at the interface between PLP droplets and RRP-RNA droplets. The angle between tangent (a) and tangent (c) was taken as the PLP contact angle ($\theta_{PLP}$). The angle between tangent (b) and tangent (c) was taken as the RRP contact angle ($\theta_{RRP}$). The contact angle values obtained from several condensates were averaged (see the "Statistics and reproducibility" section).

**Turbidity measurements**. FUS$^{RGG3}$ and poly(rU) mixtures were prepared at a fixed FUS$^{RGG3}$ concentration and variable poly(rU) concentrations in a buffer containing 25 mM Tris-HCl, 150 mM NaCl and 20 mM DTT (pH 7.5). Sample absorbance at 350 nm was measured using a spectrophotometer (Nanodrop oneC UV–Vis) with an optical path length of 1 mm. A gradual poly(rU) titration was used to record the turbidity data. Turbidity for FUS$^{RGG3}$ and poly(P) mixtures was obtained in a similar manner.

**Fluid interface simulation**. To explore the effect of surface tension on multi-phase coexistence, we used a fluid interface modeling tool (Surface Evolver v2.70)[71]. Briefly, two volumes of distinct liquids are created. Each interface is given a specific value of interfacial tension (see Figs. 4d and 6, main-text). The algorithm minimizes the total surface energy of the system using the gradient descent method[71]. As a control, we simulated the interfacial evolution of a cube of liquid, the minimization resulted in the transformation of the cube to a sphere[67] [the minimum surface energy geometry] (Fig. S22a). Throughout the minimization steps, the volumes of the two liquids were kept constant.

**Fluorescence correlation spectroscopy**. Samples containing 50 nM of Alexa488-labeled PLP were injected into a Tween-coated (Tween20) 25 mm × 75 mm × 0.1 mm custom-made flow chamber and loaded onto the microscope stage (Lumicks, C-trap) equipped with a single-photon Avalanche photodiode (sAPD). Measurements of the photon arrival times were acquired at a 100 MHz sampling rate by performing a point scan in the sample away from the glass surface. The excitation power was kept at a minimum to avoid photobleaching of the fluorophores. Each point scan was curated over 5 min. For each sample [PLP and PLP + poly(rU)], five point-scans were obtained and analyzed as follows. For each point scan, the autocorrelation function was calculated for different lag times using the

pycorrelate python library (version 0.2.1, see documentation at https://pypi.org/project/pycorrelate/#description). Five autocorrelation curves were averaged for each sample and plotted for comparison.

**Stability assay for multiphasic condensates.** Before sample preparation, FUS[PLP] was buffer exchanged in the same way as mentioned in the state-diagram analyses section. The co-existing droplet sample was prepared by mixing all the three components in a test tube (FUS[PLP], [RGRGG]$_5$, poly(rU) RNA) at the concentrations mentioned in the appropriate figure legend (Fig. S18). Approximately 500 nM of Alexa488-labeled FUS[PLP] and Alexa594-labeled RRP were used for fluorescence microscopy. The order of addition during sample preparation was buffer, FUS[PLP], [RGRGG]$_5$, fluorescent probes, and poly(rU) RNA. Approximately 5 μL volume of prepared sample was placed inside the tween20-coated Nunc Lab-Tek Chambered Cover glass and imaged using the Zeiss LSM710 laser scanning confocal microscope. The sample was covered with 100–200 μL of Fluorinert[TM] FC-770 (Sigma-Aldrich), which is a highly inert liquid and completely immiscible with water. FC-770 layer on the top of the sample helps in avoiding sample drying and preserving the sample for days.

**RNase effect on multiphasic condensates.** Before sample preparation, FUS[PLP] was buffer exchanged in the same way as mentioned in the state-diagram analyses section. The co-existing droplet sample was prepared by mixing all the three components in a test tube (FUS[PLP], [RGRGG]$_5$, rU40 RNA) at the concentrations mentioned in the appropriate figure legend (Fig. 3i, j). Approximately 500 nM of Alexa488-labeled FUS[PLP] and Alexa594-labeled [RGRGG]$_5$ were used for fluorescence microscopy. RNase-A (Thermo Scientific) was added 16% by volume to a final concentration of 1.6 mg/ml. A similar multi-phasic sample was prepared at the same time point to which buffer was added instead of RNase-A (16% by volume). Both samples, with and without RNase-A, were placed inside Tween-coated (Tween20) 25 mm × 75 mm × 0.1 mm custom-made flow chamber and sealed. Time-lapse images for both the samples were acquired using a confocal microscope (Lumicks, C-trap) for comparison.

**Statistics and reproducibility.** A two-tailed t-test was used for statistical analysis. **** represents a p-value < 0.0001, *** represents a p-value between 0.0001 and 0.001 and no star represents a p-value > 0.05. The number of droplets (n) analyzed in various figures is mentioned below.
Figure 1b, c: $n = 60$ for partition and $n = 3$ for diffusion coefficient. Figure 1k: $n = 100$ for EWS[PLP] partition, $n = 50$ for FUS[PLP] partition and $n = 75$ for BRG1[LCD] partition. Figure 4c: $n = 25$ for contact angle measurements of each droplet type ($\theta_{PLP}$ and $\theta_{RRP}$). Figure 5b: $n = 60$ for partition and $n = 3$ for diffusion coefficient measurements. All statistical measurements were done on the same sample for each distinct experimental condition. For reproducibility, all the main-text figures (Figs. 2a, c, 3a, c, 3e, f, 3h–j, 4b, 5f, g) that contain microscopy images are representative of two independent sample replicates. For Figs. 1g–i & 5c in the main-text, the reported images are representative of a large set of images from different spots in the same sample. For FRAP measurements (Figs. S3, S23), the microscopy images are representative of at least three FRAP events from different spots in the same sample. For the state diagram measurements (Figs. 1a, 5a, S1, S7, S25a), the transition points between the mixed state (no LLPS) and the phase separation state (LLPS) were reproduced twice for each system.

**Data processing software.** Excel 2016 was used for partition calculations, MATLAB (R2018a) was used for FRAP analysis and statistical analysis. Fiji-ImageJ[88] (version 1.52p) was used for image processing. OriginPro (2018b) was used for Graphing. Adobe Illustrator CC 2019 (v23.0) was used for the figure assembly and production. ZEN (blue, v2.3) was used for image recording/processing using Zeiss Primovert microscope. Bluelake (v1.6.11) was used for image recording and processing using Lumicks C-Trap microscope. Fluid-interface modeling was performed using Surface Evolver (v2.70). MD simulation was performed using HOOMD-blue (2.7.0). MD visualization was done using VMD (v1.9.4) and OVITO (v3.2.0).

**MD simulation.** In this study, we have employed a single residue/base resolution coarse-grained polyelectrolyte model for protein and RNA chains. For the amino acids, we employ the same coarse-grained parameters as have been employed by Dignon et al.[89] to study the phase behavior of intrinsically disordered proteins. The potential energy function contains bonded, electrostatic, and short-range pairwise interaction terms. Bonded interactions are modeled using a harmonic potential $k_r(r - r_0)^2$ with a spring constant $k_r = 10 \, kJ/\mathring{A}^2$ and an equilibrium bond length of $r_0 = 3.8 \, \mathring{A}$. Electrostatic interactions are modeled using a Columbic term with Debye-Hückel electrostatic screening to account for salt concentration, having the functional form:

$$E_{ij}(r) = \frac{q_i q_j}{4\pi D r} \exp\left(-\frac{r}{\kappa}\right) \qquad (3)$$

where $\kappa$ is the Debye screening length and $D = 80$, is the dielectric constant of the solvent (water). We set the Debye screening length $\kappa = 0.1$, which corresponds to

~100 mM salt concentration at room temperature. The RNA chain is modeled as a one bead per nucleotide model compatible with the protein model with the only difference being the addition of a harmonic angular term $k_\theta(\theta - \theta_0)^2$ to model the stiffness of the RNA chains, where spring constant $k_\theta = 1.0 \, kJ$ and equilibrium angle $\theta_0 = 1.78$ rads.

The initial system configurations were generated by placing the protein and RNA chains randomly in the simulation box at concentrations corresponding to the experimental conditions. The system was energy-minimized with an energy tolerance of $10^{-7}$ kJ/mole and force tolerance of $10^{-7}$ kJ/mole-Å. The system was then equilibrated in the canonical constant Number of particles, Pressure, and Temperature (NPT) ensemble at 298 K and 1 atm using the Nose-Hoover thermostat and barostat with a coupling time constant of 1 ps and 10 ps respectively. The equilibrated system was further stimulated in the canonical NVT ensemble at 298 K using the Langevin thermostat with a friction coefficient of $\gamma = 0.01$. MD runs were performed on graphical processing units (GPUs) using the HOOMD-blue package v2.7.0[90,91]. The equilibrium run was performed for 10 ns with a time step of 0.01 ps and the production run was performed for an additional 10 ns. The surface-recruitment simulations shown in Fig. 1f (main-text) and S27 were performed in two stages. In the first stage, the condensates under low RNA and high RNA conditions were generated using the procedure above. In the second stage, PLP chains were randomly placed in the simulation box and the system was then equilibrated in the constant NVT ensemble at 298 K for 10 ns.

**Reporting summary.** Further information on research design is available in the Nature Research Reporting Summary linked to this article.

## Data availability
The data supporting the findings of this study are available from the corresponding authors upon reasonable request. Source data are provided with this paper.

## Code availability
We used the publicly available HOOMD-blue package (v2.7.0)[90] for molecular dynamics simulations. Fluid-interface modeling was done using the freely available software SurfaceEvolver (v2.70)[71]. Custom codes for the analysis and production of the results reported in this paper can be made available from the corresponding author upon reasonable request.

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

## Acknowledgements
The authors gratefully acknowledge UB north campus confocal imaging facility and its director, Mr. Alan Siegel for helpful assistance. The authors acknowledge Ms. Liz-Audrey Djomnang Kounatse for her help with FUS^RGG3-poly(rU) phase diagram. The authors also acknowledge helpful discussions with Dr. George Thurston, and Dr. Mahdi M. Moosa at various stages of manuscript preparation. We gratefully acknowledge support for this work from University at Buffalo, SUNY, College of Arts and Sciences to P.R.B. and funding from the National Institute of General Medical Sciences (NIGMS) of the National Institutes of Health (R35 GM138186) to P.R.B. D.A.P. acknowledges financial support from Iowa State University. D.A.P. also acknowledges NSF Extreme Science and Engineering Discovery Environment allocation on Bridges graphical processing unit machine at the service provider through Allocation CTS190023.

## Author contributions
P.R.B. and T.K. conceived the idea and designed the experiments. T.K. performed the experiments and analyzed the data with help from P.R.B. and I.A. R.B.D. expressed and purified recombinant proteins and performed their fluorescent labeling. I.A. performed the fluid-interface modeling simulations. M.R. and D.A.P. designed and performed the MD simulations. P.R.B., T.K., and I.A. wrote the manuscript with input from R.B.D., M.R., and D.A.P.

## Competing interests
The authors declare no competing interests.
