## [Peer Review File · Nature Communications]

REVIEWER COMMENTS

Reviewer #1 (Remarks to the Author):

Please see the detailed report that is attached here as a PDF.

This manuscript by Kaur et al., represents a tour de force contribution from the Banerjee lab, supported by computational work from the Potoyan lab. Here, the authors address the issue of compositional specificity in the simplest facsimiles of multicomponent systems that approximate bona fide biomolecular condensates. The rules that emerge from this work are going to be exceptionally useful for the field and they integrate well with findings from the soft matter physics literature as well. Without a doubt, this work deserves to be published and it is likely to become something of a citation classic given the range of questions that it answers and the elegant methods that are brought to bear in answering these questions. From a biological perspective, the data in this work provide a physico-chemical impetus for thinking about and formulating testable hypotheses regarding the spatial and compositional regulation of multicomponent biomolecular condensates. Work like this starts to move us in the direction of developing a clearer understanding of the extent to which sequence-encoded interactions can provide chemostatic control over condensates, and possibly their functions.

Overall, this MS has numerous positives that outnumber and outweigh the issues that arose during careful reading. That said, there are some key issues that need close scrutiny. Of particular relevance is the assertion (without conclusive evidence) that surface-mediated interactions almost exclusively control the compositions, the spatial organization, and the extent of mixing / wetting of condensates. Additionally, the reliance on the micellar metaphor has issues. And finally, there are errors of omission with regard to key papers in the literature that should be remedied. Clearly, some significant and targeted revisions are essential. These revisions do not (for the most part) require new experiments or simulations, although they might require re-analysis of the data, questioning of certain assertions, and clarifying key matters in the text. The issues that came up during my reading are inventoried below. Some (perhaps all) of these will be viewed as being rather persnickety, but the hope is that these will be seen as constructive suggestions to be taken into consideration in a suitably revised version.

Technical points

1. The most important findings from this work, which are also highly relevant, are beautifully summarized in the following sentences of the introduction "Within the PLD-RLDRNA ternary system, RNA-RLD interactions dominate over PLD-RLD interactions, leading to an RNA-induced demixing of PLD-RLD condensates into stable coexisting phases (homotypic PLD condensates and heterotypic RLD-RNA condensates). The topology of these biphasic condensates (non-engulfing/ partial engulfing/ complete engulfing) is determined by the RNA-to-RLD stoichiometry and the hierarchy of intermolecular interactions, providing a glimpse of the broad range of multiphasic patterns that are accessible to these condensates." It would be beneficial to capture these statements and convey them in the abstract, which does not, at this juncture, lay out the findings in their clearest form.
2. There is the repeated assertion that the interfaces are between pairs of liquids. Is this a hypothesis or are there data to support this assertion? This is important because the concept of surface tension applies only to liquid-liquid interfaces whereas the term surface energy is much more general.
3. Among nucleic acid aficionados, it is accepted that homopolymeric RNA molecules are referred to as poly-r[A,U,G,C]. Here, the "r" helps distinguish RNA from single-stranded DNA where the convention would be to use poly-d[A,T,G,C]. While these rigorous conventions are rarely used, it should be used because soon we will start seeing lots of papers showcasing the effects of single-stranded DNA molecules and this will require being able to distinguish poly-rA from poly-dA for example.
4. Why did the authors choose poly-rU? As shown by Boeynaems et al., the identity of the nucleobase does matter and a thread left hanging is how the elegant results presented by Kaur et al., are likely to be impacted by details regarding the RNA.
5. The authors refer to the concentration threshold for phase separation of the PLD as and they

propose that the RLD decreases this concentration monotonically. This, in the parlance of polyphasic linkage, will imply that RLD acts primarily as a ligand that binds to the PLD preferentially in the dense phase. Are the authors arguing for this phenomenon and if so, is this because of the reasonable assumption (supported by their own observations) that the RLD (at least in the concentration regimes investigated here) does not phase separate on its own? This is clearly the case, and therefore it is important to place the findings in the context of the polyphasic linkage formalism. This formalism, first introduced by Wyman and Gill (1980) has been made tractable by the recent work of Ruff et al. (2020) – please see: <https://www.biorxiv.org/content/10.1101/2020.08.15.252346v1>. Please use this formalism to quantify how varies with [RLD]. In doing so, it would help to analyze the results in terms of the concentration of aromatic stickers in the PLD and concentration of Arg residues in the RLD. This is relevant because for polymers, molar units become highly misleading. To remedy this issue, one should either use volume fractions or since these are associative polymers, it makes more sense to use molarity or concentrations of stickers. The authors use a mixture of units, molarity for protein concentrations and wt / wt ratios for protein : protein or protein : RNA ratios. These are relative mass fractions, but this can also become confusing and it is particularly problematic to mix units as is done in this work.

6. The work of Ruff et al., cited above, also raises serious questions about the insights to be gained from partition coefficient measurements. In fact, a reader might be misled into thinking that the condensates are becoming more concentrated in PLDs as the RLD concentration increases. Condensates do not have infinite free volume and instead, the increased PC is most likely due to the lowering of and not due to an increase in the dense phase concentration of the PLD. This is really important and given the existence of measurements of vs. RLD concentration, the utility of the PC measurements is unclear.

7. The so-called monotonic vs. “highly non-monotonic” behaviors alluded to by the authors are readily explained in terms of two recent advances, one that builds on the pioneering work of Wyman & Gill (see Ruff et al., referenced above) and the other that highlights the unique issues that emerge in thinking about phase diagrams for multicomponent systems. Simply stated, while the RLD is a ligand for the PLD the RLD-RNA system is under the control, primarily, of heterotypic interactions. This engenders closed loop phase diagrams, which will lead to reentrant phase behavior. Please see: <https://doi.org/10.1371/journal.pcbi.1007028>, <https://doi.org/10.1038/s41586-020-2256-2>, and <https://arxiv.org/abs/1910.11193>. Cast in the light of these advances, the reentrant behavior is not surprising and should not be referred to as “highly non-monotonic” that is cLLPS PLD cLLPS PLD cLLPS PLD cLLPS PLD to be contrasted with the “monotonic” behavior in the PLD-RLD mixtures. One is a manifestation of the RLD being a ligand that binds preferentially to the PLD in its dense phase and the other is a manifestation of the dominant contributions of heterotypic interactions to the phase behavior in RLD-RNA mixtures. It is quite remarkable that the scenarios articulated by Ruff et al., are borne out in the current work. This is really neat.

8. To the point raised above, there are three possible inferences to be drawn from the data in Figure S4. The PC for RNA in PLD condensates would equal one, be less than one or, greater than one. Which of these is true? Preferential exclusion with a $PC < 1$ would be intuitive, but this should then show up as linkage effect and hence influence. Alternatively, a $PC \geq 1$ would be intuitive to some if $PC = 1$, agnostic interactions or $PC > 1$ highlighting the passive partitioning of RNA into a region where the local concentration of weakly interacting sites is rather high. This would still be consistent with the FCS data in S4. Therefore, it would be clarifying and helpful to measure the PCs for RNA into PLD condensates.

9. I have serious problems with referring to condensates as being micellar. First, micelles are known as pseudo-phases. Second, micelles will have a fixed size and this size as well as the molecularity of the micelle will be governed by molecular dimensions of the components. There is no way a micellar system can make micron-scale condensates. Third, do the sizes of condensates increase with concentration or do the sizes stay fixed and the numbers of condensates increase? If this is the case, these are micellar condensates and the phase separation is an exemplar of micro-phase separation. The Chilkoti lab has shown how this can be realized in block co-polypeptide systems. Fourth, if the simulations are suggestive of micellar architectures, then the question is these are indeed bona fide micelles and if yes, how does one justify micelles growing to be spherical condensates and not forming

cylinders, toroids, or bicontinuous phases? It is more than likely that the observed micellar features are largely a consequence of the serious finite size artefacts that prevail in the simulations.

10. "To validate the micellar condensate formation in our system, we turn to molecular dynamics simulations...". This phrase is problematic on so many levels. First, see the concerns raised above. Second, one should never use the term validate. It conveys the impression of certitude. The better alternative is to state that "in order to provide a molecular-level understanding of..."

11. The cartoon representations in Figures 1e, 4e, and 5d seem to convey the impression of essentially vesicular structures. Almost certainly there will be molecular scale or larger scale roughness on the surface of the condensates, and this would be indicative of a mixture (favoring one species over the other) of compositions at the surface. If that is not the case, then the interfacial tensions would be rather high, and all measurements to date place these numbers at being 6-7 orders of magnitude smaller than that of the air-water interface. This is important because a casual reader will walk away with the wrong impression.

12. Figure 1g is important, and in the context of what is now known, these data suggest that condensate formation, at least under the conditions investigated, is probably under the exclusive control of heterotypic interactions in the RLD-RNA system. If this were not the case, one would not observe a one-to-one correspondence between the bulk ratios of RNA cLLPS PLD to RLD on the partitioning of PLDs into these condensates. This point has to be emphasized more thoroughly.

13. The authors, based on simulations in systems that are likely plagued by finite size issues, propose, essentially assert that the sites on the surfaces of RLD-RNA condensates are the relevant species. Why should this be true? Figure 1j would appear to offer experimental data in support of the contention, but the level of surface enrichment is a) rather modest, b) system-specific, and c) does not account for the relative differences in the number of sites at the interface vs. the interior, which would scale as $R^2/3$ where R is the radius of the condensate. And the system specificity might arise due to comparisons being made at nonequivalent quench depths and / or not normalizing the effects in terms of the numbers / concentrations of aromatic stickers. This raises serious questions about Figure 1k as well and I fear that the surface-centric narrative is being pushed largely by the features identified in simulations plagued by finite size considerations. In fact, pure surface binding cannot enable the condensate switching observed in this work.

14. The effect of condensate dissolution by RNA, for which the authors have unique data sets, should be analyzed using the polyphasic linkage formalism. The protocol prescribed by Ruff et al., would be helpful as would the work of Posey et al. (see: <https://www.jbc.org/content/293/10/3734>).

15. The term client is a misnomer given the observations of preferential binding, dissolution, and sorting. By the definitions of Rosen and colleagues, a client is simply a molecule that has a $PC > 1$. What the authors are reporting is really a ligand-aided sorting mechanism via polyphasic linkage followed by contact without wetting of unmixed condensates. This behavior has been reported by the Gladfelter lab in a series of reports dating back to 2013. These papers have been ignored in the narrative, and this error of omission has to be remedied.

16. When the authors introduce the nomenclature for comparing interfacial tensions, it is unclear if they are referring to a binary mixture. This is what the A, B, and C designations appear to suggest. If so, the relevance of the inequalities is lost on me for a ternary system. Also, the authors do not once mention Neuman's inequality nor do they connect with the copious literature on this subject.

17. The schematic in Figure 4a seems to be straight out the review published by Shin and Brangwynne. If the authors wish to use this graphic, then they should do so with attribution, the permission of the authors, and permission from Science magazine. Otherwise, they should revise the graphic and still cite the Shin and Brangwynne paper in the figure caption.

18. Equation (1) introduces three interfacial tensions viz., g_{RLD} , g_{PLD} , and $g_{PLD-RLD}$. The narrative would suggest that g_{PLD} is the interfacial tension that describe the interface formed between PLD condensates and a dispersed phase that is solvent-rich and PLD-deficient. Likewise, $g_{PLD-RLD}$ should be the interfacial tension at the interface between PLD-RLD condensates and a dilute solution comprising PLD and RLD molecules. So, what then is g_{RLD} ? Afterall, the RLD does not form condensates with a defined phase boundary. So, this equation is probably incorrect. This impacts all of the analysis presented in Figure 4 and it should be clarified and revisited.

19. With regard to the differences between Lys and Arg, the recent work of Greig et al., is very much

in the spirit of the findings reported here, and they have been ignored. This should be remedied. Please see: <https://doi.org/10.1016/j.molcel.2020.01.025>. The authors are advancing the importance of sequence-encoded interactions as determinants of condensate regulation, morphology, and spatial organization. This points to cellular control over the molecular grammar and accordingly, it is important to ensure that preceding contributions not be ignored or given short shrift. Apportioning due credit does not take away from the current work, it instead buttresses its importance.

20. With all due respect, the data stand on their own without the MD simulations. To assert otherwise is misleading. The key insight that emerges from the simulations is surfacecentric view, and this should, at this juncture be framed more as a hypothesis rather than fact because this is where the data are particularly unclear. If the authors wish to assert otherwise, then it would help immensely to perform computational interrogations that lead to predictions of the relative interfacial tensions for different condensates whilst taking care to guard against finite size artefacts. This is a tall order, and in my view, it is beyond the scope of the current work. Instead, I propose toning down the pronouncements regarding the predictive utility of the MD simulations. These simulations provide a picture that may or may not be true and their accuracy remains undetermined at this time.

21. The authors propose that fuzzy interactions, non-stoichiometric interactions are the central determinants of the numerous interesting observations they present in this work. What makes interactions fuzzy and how does one quantify such fuzzy interactions? It would have to be through the device of distribution functions. If so, then are the effects of interactions quantified using pair distribution functions in liquids to be referred to as being fuzzy? I shouldn't think so and this nomenclature is not helpful because we end up with a conflation of thermodynamics and dynamics concepts that are not easy to follow. Additionally, the assertion regarding interactions being non-stoichiometric is a non-quantitative one. As has been noted recently, the numbers (valence) of stickers and the modulation of their interactions by spacers directly influence the driving forces for phase separation aided percolation transitions. These interactions can give rise to stoichiometric assemblies or non-stoichiometric assemblies that are highly networked. This would imply that it is misleading to refer to interactions as being non-stoichiometric. If the authors wish to use this verbiage, then they should explain exactly what they mean by this usage.

Semantic and conceptual issues

1. There is the repeating usage of the phrase "networks of overlapping interactions". This is confusing. What does the term "overlapping" mean in this context? Please clarify and / or rephrase. Close reading suggests that the authors might be referring to concentration (or stoichiometry) dependent competition between competing interactions (maybe).

2. Please use the more widely accepted term biomolecular condensates instead of biocondensates. And, given that BMC is an abbreviation that has been co-opted for other topics in biology, it is best to adopt the current working approach and use the term condensates if one wants to avoid using the term biomolecular condensates due to space considerations. Please do not use the BMC abbreviation. This makes things confusing.

3. The abbreviation RLD is also non-standard and it does not fit as an abbreviation either for R-rich polypeptide or for R-rich low complex domain. The better approach would be something like RRD or RRP for R-rich domain or R-rich polypeptide.

4. Does the term "topology" best describe the mesoscale organization or structure of condensates?

5. Please replace liquid-liquid phase transition with one of the following: (i) phase transition or (ii) liquid-liquid phase separation or (iii) phase separation aided percolation.

6. In motivating one's work, it is common practice to set things up with the typical declaration that something is "largely unexplored" or "poorly understood" or "controversial" etc. The authors take a similar approach and it does disservice to what has come before and the very real possibility that the current work builds on what has come before. Therefore, simply state the phenomenon one wishes to explain or describe and state something like "Here, we use a tractable, minimalist ternary system to ..." – as in state what the paper is about without anointing it as being novel or poorly understood or hitherto unexplored. If one broadens one's horizons, it becomes patently obvious that the concepts have been very well established in the context of synthetic systems that undergo complex coacervation. The work stands on its own. No need for unnecessary edification.

7. The authors list "n rich" electron systems Y/P/Q/G/S as being enriched in LCDs. If the authors are referring to PLDs, then Pro definitely is not prominent in these sequences. Perhaps the authors mean F for Phe. Also, Asn shows up more frequently in bona fide PLDs than Gln. Please see the work of Alberti et al., published in Cell and Halfmann et al., published in Molecular Cell over a decade ago. Finally, it is not clear that Q, G, and S are best described as being "n rich" systems. This nomenclature has been suggested by Vernon et al., but it is disputable. From a zeroth order perspective, it would be best to refer to these residues as polar residues capable of forming hydrogen bonds.
8. In what sense are R-rich LCDs (please change this phrase) "promiscuous RNA binders"?
9. The term "promiscuous" is thrown around somewhat loosely. In this context, it is misleading, and it often leads to the trope of IDR-mediated interactions as being nonspecific. In the context of the current work, the authors wish to imply that there are likely to be competing interaction networks and that they intend to investigate the consequences of these competing effects.
10. The italicized phrases "monotonically" and "highly non-monotonic" should be nonitalicized and are actually misleading because it is a tautology that if the "RLD" is in vast excess when compared to the PLD, then even the PLD-RLD system will show reentrant behavior or the RLD will end up wetting the PLD. The concentration regimes where this might be realized have not been explored in the current work and therefore one should be cautious about making these assertions.
11. The word uptaken should be taken up and there is one place where the authors refer to deferential partitioning, which should be differential partitioning.
12. In many of the figure captions, the authors write that "each type of droplets was...". This should be fixed to "each type of droplet was...".
13. Please note that

Reviewer #2 (Remarks to the Author):

This is one of the best studies I read recently. This work adds significantly to the field and will have a noticeable impact. It is nicely written and contains an impressive amount of convincing data. The presented statistical analysis is appropriate and valid. The level of details provided in the manuscript is sufficient to reproduce the work. This work is novel and convincing, and data described will be of interest to others in the community and the wider field.

Reviewer #3 (Remarks to the Author):

There is currently great interest in understanding molecular mechanisms of liquid-liquid phase separation of protein/RNA mixtures. In this manuscript, Kaur et al. have examined a well-defined ternary system consisting of a low-complexity (prion-like) polypeptide (PLD), Arg-rich Polypeptide (RLD) and RNA. Analysis of experimental data, together with the results of MD simulations, allowed the authors to explain, in molecular terms, the topology of these three-component condensates and the mechanisms that control this topology. Overall, the manuscript is very interesting and clearly written, and the main conclusions well supported by experimental data. This work represents a significant advance in our understanding of multi-component condensates.

Specific points:

1. My only significant concern is the lack of a more comprehensive discussion of the present findings in a biological context. Specifically, proteins driving liquid-liquid phase separation in cells typically contain both low-complexity and RNA-binding domains. It would be helpful if the authors could discuss how their findings using the isolated domains (or models thereof) alone could explain the behavior of systems containing full-length proteins.

Minor points:

1. Fig 1: From the microscopic images shown in panels g-i it appears that the proportion of protein that accumulates within the droplets' interior is substantially larger for BRG1 LCD than for two other proteins studied. Some explanation of this quantitative differences should be provided.
2. Fig. 2a and text in the last paragraph on p. 6: The authors write "The dissolution of PLD-RLD droplets is preceded by a change in their color from yellow (RLD+PLD) to green (PLD), indication that RLD is leaving these droplets...". Shouldn't it be ...from yellow (RLD+PLD) to red, indicating that PLD is leaving these droplets...?
3. Fig. S3a: Why the size of PLD droplets alone is much larger than those in the presence of RLD? This needs to be clarified.
4. Fig. 3c and text on p. 7: In contrast to what the authors state in the text, small RLD (red)PLD droplets appear to be also present without attachment to the surface of PLD (green) droplets. This should be clarified.
5. p. 9, second line: It should be Fig. 4c (not (Fig. c)).

Reviewers' comments are colored in **black**, the authors' response is colored in **blue**. The major changes made are highlighted in **yellow** in the revised manuscript and listed in the context of individual comments of the reviewers below.

Reviewer #1 (Remarks to the Author):

This manuscript by Kaur et al., represents a tour de force contribution from the Banerjee lab, supported by computational work from the Potoyan lab. Here, the authors address the issue of compositional specificity in the simplest facsimiles of multicomponent systems that approximate *bona fide* biomolecular condensates. The rules that emerge from this work are going to be exceptionally useful for the field and they integrate well with findings from the soft matter physics literature as well. Without a doubt, this work deserves to be published and it is likely to become something of a citation classic given the range of questions that it answers and the elegant methods that are brought to bear in answering these questions. From a biological perspective, the data in this work provide a physico-chemical impetus for thinking about and formulating testable hypotheses regarding the spatial and compositional regulation of multicomponent biomolecular condensates. Work like this starts to move us in the direction of developing a clearer understanding of the extent to which sequence-encoded interactions can provide chemostatic control over condensates, and possibly their functions. Overall, this MS has numerous positives that outnumber and outweigh the issues that arose during careful reading. That said, there are some key issues that need close scrutiny. Of particular relevance is the assertion (without conclusive evidence) that surface-mediated interactions almost exclusively control the compositions, the spatial organization, and the extent of mixing / wetting of condensates. Additionally, the reliance on the micellar metaphor has issues. And finally, there are errors of omission with regard to key papers in the literature that should be remedied. Clearly, some significant and targeted revisions are essential. These revisions do not (for the most part) require new experiments or simulations, although they might require re-analysis of the data, questioning of certain assertions, and clarifying key matters in the text. The issues that came up during my reading are inventoried below. Some (perhaps all) of these will be viewed as being rather persnickety, but the hope is that these will be seen as constructive suggestions to be taken into consideration in a suitably revised version.

We thank the reviewer for his/her kind words, constructive comments, and suggestions. We are glad to learn that the reviewer has found this work to be important for the field. We deeply appreciate the care and effort of the reviewer in helping us to improve our manuscript.

Technical points

1. The most important findings from this work, which are also highly relevant, are beautifully summarized in the following sentences of the introduction “Within the PLD-RLDRNA ternary system, RNA-RLD interactions dominate over PLD-RLD interactions, leading to an RNA-induced demixing of PLD-RLD condensates into stable coexisting phases (homotypic PLD condensates and heterotypic RLD-RNA condensates). The topology of these biphasic condensates (non-engulfing/ partial engulfing/ complete engulfing) is determined by the RNA-to-RLD stoichiometry and the hierarchy of intermolecular interactions, providing a glimpse of the broad range of multiphasic patterns that are

accessible to these condensates.” It would be beneficial to capture these statements and convey them in the abstract, which does not, at this juncture, lay out the findings in their clearest form.

We thank the reviewer for appreciating the importance of our work. We have now revised the abstract to better capture the essence of our findings. It now reads:

“.....we show that competition between the PLP and RNA for a single shared partner, the RRP, leads to RNA-induced demixing of PLP-RRP condensates into stable coexisting phases—homotypic PLP condensates and heterotypic RRP-RNA condensates. The morphology of these biphasic condensates (non-engulfing/ partial engulfing/ complete engulfing) is determined by the RNA-to-RRP stoichiometry and the hierarchy of intermolecular interactions, providing a glimpse of the broad range of multiphasic patterns that are accessible to these condensates.”

2. There is the repeated assertion that the interfaces are between pairs of liquids. Is this a hypothesis or are there data to support this assertion? This is important because the concept of surface tension applies only to liquid-liquid interfaces whereas the term surface energy is much more general.

We thank the reviewer for raising this point. The use of surface tension terminology is adopted based on our experimental results showing stable coexisting phases of homotypic PLP condensates and heterotypic RRP-RNA condensates. Although PLP condensates do not mix with RRP-RNA condensates, both types of condensate are fluid droplets. This is because PLP droplets undergo rapid fusion with PLP droplets in contact, and similarly, RRP-RNA droplets coalesce with each other (Fig. S20). In addition to these results, we have now added a new set of experimental data showing that the RNA-induced demixing of PLP-RRP condensates into coexisting PLP condensates and RRP-RNA condensates can be reversed by the introduction of an RNA-degrading enzyme RNase-A (Fig. 3i&j). Hence, we chose the term “surface tension” to describe interactions between a pair of liquids (PLP droplets and RRP-RNA droplets), immersed in a dilute phase. Following the reviewer’s comments, we now realized that the term “interfacial tension” is more appropriate to use for a pair of liquid droplets dispersed in a 3rd liquid phase (Rowlinson and Widom, *Molecular theory of capillarity*; Evans and Wennerstrom, *The colloidal domain*). Hence, in the revised manuscript, we have used interfacial tension in the context of a liquid-liquid interface.

3. Among nucleic acid aficionados, it is accepted that homopolymeric RNA molecules are referred to as poly-r[A,U,G,C]. Here, the “r” helps distinguish RNA from single-stranded DNA where the convention would be to use poly-d[A,T,G,C]. While these rigorous conventions are rarely used, it should be used because soon we will start seeing lots of papers showcasing the effects of single-stranded DNA molecules and this will require being able to distinguish poly-rA from poly-dA for example.

We agree with the reviewer on this point and we have now referred to the poly(U) RNA as poly(rU), poly(A) RNA as poly(rA), U40 RNA as rU40, and U10 RNA as rU10.

4. Why did the authors choose poly-rU? As shown by Boeynaems et al., the identity of the nucleobase does matter and a thread left hanging is how the elegant results presented by Kaur et al., are likely to be impacted by details regarding the RNA.

The reviewer is correct to point out the role played by the RNA sequence and structure. Indeed, papers by Boeynaems et al. (Boeynaems, S. et al., 2019, *Proceedings of the National Academy of Sciences*, 116(16), 7889-7898) and our earlier work (Alshareedah, I. et al., 2019, *Journal of the American Chemical Society*, 141(37), 14593-14602) showed that varying the RNA impacts the phase behavior and condensate material properties of protein-RNA mixtures. For this reason, we have conducted experiments with poly(rA) as well as with shorter poly(rU) chain: rU40. We note that these data were reported in the original submission (see Figs. 2c and 3e&f). In addition to these results, we have now conducted new experiments and included data for the coexistence behavior of PLP condensates with RRP-RNA condensate using yeast total RNA mixture (Fig. S19). For all these RNAs, we observed stable coexistence of PLP droplets with RRP-RNA droplets. We hope that these results are sufficient to address the reviewer's concern.

5. The authors refer to the concentration threshold for phase separation of the PLD as c_{LLPS}^{PLD} and they propose that the RLD decreases this concentration monotonically. This, in the parlance of polyphasic linkage, will imply that RLD acts primarily as a ligand that binds to the PLD preferentially in the dense phase. Are the authors arguing for this phenomenon and if so, is this because of the reasonable assumption (supported by their own observations) that the RLD (at least in the concentration regimes investigated here) does not phase separate on its own? This is clearly the case, and therefore it is important to place the findings in the context of the polyphasic linkage formalism. This formalism, first introduced by Wyman and Gill (1980) has been made tractable by the recent work of Ruff et al. (2020) – please see: <https://www.biorxiv.org/content/10.1101/2020.08.15.252346v1>. Please use this formalism to quantify how c_{LLPS}^{PLD} varies with [RLD]. In doing so, it would help to analyze the results in terms of the concentration of aromatic stickers in the PLD and concentration of Arg residues in the RLD. This is relevant because for polymers, molar units become highly misleading. To remedy this issue, one should either use volume fractions or since these are associative polymers, it makes more sense to use molarity or concentrations of stickers. The authors use a mixture of units, molarity for protein concentrations and wt/wt ratios for protein:protein or protein:RNA ratios. These are relative mass fractions, but this can also become confusing and it is particularly problematic to mix units as is done in this work.

The reviewer is right; we do not observe the formation of homotypic condensates by the R-rich polypeptide in our experimentally-probed concentration regime. Since (a) RRP-PLP interactions may not be obligate heterotypic (given PLP-PLP homotypic interaction is present), and (b) PLP saturation concentration is lowered by the RRP, it is reasonable to argue that *RRP acts primarily as a ligand that binds to the PLP preferentially in the dense phase*. In the revised manuscript, we have cited these two relevant papers as suggested by the reviewer and revised our discussion as follows:

“Our analysis revealed that the phase separation of PLP is enhanced with RRP in a composition-dependent manner. Specifically, the LLPS concentration threshold for PLP (C_{LLPS}^{PLP}) decreases monotonically with increasing RRP concentration [in the absence of the RRP, $C_{LLPS}^{PLP} = C_{saturation}^{PLP} \sim 240 \mu\text{M}$; at an RRP-to-PLP ratio of 5, $C_{LLPS}^{PLP} \sim 120 \mu\text{M}$] (Fig. 1a; Figs. S1&S2) under our experimentally tested conditions (up to RRP-to-PLP ratio of 20:1). Although PLP readily undergoes homotypic condensation in the absence of RRP, the latter did not show any sign of homotypic LLPS at the concentrations used in our study. Hence, our results that the saturation concentration of PLP decreases monotonically with [RRP] are consistent with a scenario where RRP acts as a ligand that binds to the PLP preferentially in the dense phase^{52,53}.”

In the revised manuscript, we have also included a plot showing how C_{LLPS}^{PLP} varies as a function of [RRP] (Fig. S2b).

Finally, we agree that the mixing of units may lead to confusion, therefore, we have now provided the same graphs to show the concentrations in molar units of Arg in RRP and Tyr (aromatic sticker) in PLP (Figs. S2 and S7).

6. The work of Ruff et al., cited above, also raises serious questions about the insights to be gained from partition coefficient measurements. In fact, a reader might be misled into thinking that the condensates are becoming more concentrated in PLDs as the RLD concentration increases. Condensates do not have infinite free volume and instead, the increased PC is most likely due to the lowering of C_{LLPS}^{PLD} and not due to an increase in the dense phase concentration of the PLD. This is really important and given the existence of measurements of C_{LLPS}^{PLD} vs. RLD concentration, the utility of the PC measurements is unclear.

The plot of C_{LLPS}^{PLP} vs [RRP] (shown in Fig. S2 in the revised manuscript) indeed shows that the PLP saturation concentration (which is the same as C_{Dilute}^{PLP}) decreases substantially with increasing [RRP], which can result in increasing the PC ($K = C_{Dense}^{PLP}/C_{Dilute}^{PLP}$) without any substantial changes to the dense phase concentration of the PLP (C_{Dense}^{PLP}). On the other hand, we also found that PLP diffusion substantially slows down with increasing RRP (Fig. 1c). The state diagram is shown in Fig. 1a (also see Fig. S2) which hints that the concentration of the dilute phase is decreasing which would lead to an increase in the partition coefficient as observed. The diffusion data may indicate that the density in the dense phase is increasing, which leads to slower diffusion and a larger partition coefficient. However, without further quantification of the dense and dilute phase concentrations, which would require further experiments and careful calibration methods, we opted not to quantitatively comment on the implications of the partition coefficient data in the original submission. Following the reviewer’s suggestions, we have revised the manuscript and clearly stated the possibility that an increase in the PLP PC in presence of RRP could be a manifestation of RRP-aided PLP phase separation, which should remove any confusion among the readers regarding the changes in [PLP] in these condensates. The revised text now reads:

In addition to the altered phase behavior of PLP due to the presence of RRP, we observe that the PLP partition coefficient ($K = C_{Dense}^{PLP}/C_{Dilute}^{PLP}$) increases with [RRP]. This is a direct manifestation of the lowering of PLP saturation concentration by the RRP. Simultaneously, fluorescence recovery after photobleaching (FRAP) assays indicate that the PLP mobility (D_{app}) in the dense phase decreases with increasing [RRP] (Fig. 1b&c, Figs. S3&S4). These results are consistent with previous literature reports^{45,49} and indicate that RRP prefer binding to the PLP in the dense phase^{52,53}, thereby enhancing PLP phase separation and impacting the condensate dynamics (Fig. 1a-c).

7. The so-called monotonic vs. “highly non-monotonic” behaviors alluded to by the authors are readily explained in terms of two recent advances, one that builds on the pioneering work of Wyman & Gill (see Ruff et al., referenced above) and the other that highlights the unique issues that emerge in thinking about phase diagrams for multicomponent systems. Simply stated, while the RLD is a ligand for the PLD the RLD-RNA system is under the control, primarily, of heterotypic interactions. This engenders closed loop phase diagrams, which will lead to reentrant phase behavior. Please see: <https://doi.org/10.1371/journal.pcbi.1007028>, <https://doi.org/10.1038/s41586-020-2256-2>, and <https://arxiv.org/abs/1910.11193>. Cast in the light of these advances, the reentrant behavior is not surprising and should not be referred to as “highly non-monotonic” that is to be contrasted with the “monotonic” behavior in the PLD-RLD mixtures. One is a manifestation of the RLD being a ligand that binds preferentially to the PLD in its dense phase and the other is a manifestation of the dominant contributions of heterotypic interactions to the phase behavior in RLD-RNA mixtures. It is quite remarkable that the scenarios articulated by Ruff et al., are borne out in the current work. This is really neat.

We acknowledge the reviewer’s point here. The reviewer is correct in saying that the contrasting phase behaviors of RRP-PLP and RRP-RNA mixtures can be explained by the recent developments in the field (abovementioned references) and our work on reentrant phase transition (Banerjee et al., 2017, *Angew Chem Int Ed Engl.* 56(38):11354-11359; Alshareedah et al. 2019, *J. Am. Chem. Soc.* 141, 37, 14593-14602). However, we opted to contrast these two systems to alert the unfamiliar readers to these differences in phase behavior of multicomponent mixtures. It is true that to the readers who are versed with the thermodynamics of multicomponent mixtures, the differences may be expected given the wealth of literature in polymer systems and globular proteins that span more than 3 decades. But to the general biophysics audience, we feel that it would be helpful to point out these differences by describing them in simple terms such as monotonic vs. non-monotonic. Therefore, contrasting the shape of these state diagrams simply asserts that multi-component systems have a rich phase behavior as compared to single-component systems (as the three papers suggested by the reviewer clearly document). To address the reviewer’s concern, we have now added two sentences stating that these differences are expected in light of the recent theoretical and experimental advancements and cited the appropriate literature (Ruff et al., *BioRxiv* <https://doi.org/10.1101/2020.08.15.252346>; Choi, J. M.,

Dar, F., & Pappu, R. V. (2019). PLOS Computational Biology, 15(10), e1007028; Riback, J. A. et al., (2020). Nature, 581(7807), 209-214; Nandi et al., arXiv preprint arXiv:1910.11193 2019; Nguyen, T. T., Rouzina, I., & Shklovskii, B. I. (2000). The Journal of chemical physics, 112(5), 2562-2568; Nguyen, T. T., & Shklovskii, B. I. (2001). The Journal of Chemical Physics, 115(15), 7298-7308; Zhang, R., & Shklovskii, B. I. (2005). Physica A: Statistical Mechanics and its Applications, 352(1), 216-238; Grosberg, A. Y., Nguyen, T. T., & Shklovskii, B. I. (2002). Reviews of modern physics, 74(2), 329).

The revised statement now reads: “*Similar to RRP-PLP mixtures, RRP-RNA mixtures display a composition-dependent phase behavior. However, unlike RRP-PLP mixtures, their phase behavior is non-monotonic, wherein the two-phase regime is only stabilized within a small window of mixture compositions (Fig. 1d). Such composition-dependent phase separation is a hallmark of multi-component systems and is usually referred to as reentrant phase transition^{28,50,54}. The observed difference in the phase behavior of RRP-RNA and RRP-PLP mixtures is expected in light of recent theoretical developments which indicate that multicomponent mixtures with obligate heterotypic interactions often display reentrant phase behavior⁵⁵⁻⁶¹.*”

8. To the point raised above, there are three possible inferences to be drawn from the data in Figure S4. The PC for RNA in PLD condensates would equal one, be less than one or, greater than one. Which of these is true? Preferential exclusion with a PC < 1 would be intuitive, but this should then show up as linkage effect and hence influence C_{LLPS}^{PLD} . Alternatively, a PC ≥ 1 would be intuitive to some if PC = 1, agnostic interactions or PC > 1 highlighting the passive partitioning of RNA into a region where the local concentration of weakly interacting sites is rather high. This would still be consistent with the FCS data in S4. Therefore, it would be clarifying and helpful to measure the PCs for RNA into PLD condensates.

We have now measured the partition of fluorescently labeled RNA (rU10) within PLP condensates in the absence and presence of poly(rU) RNA, which suggests that $PC_{RNA} \leq 1.0$ (Fig. S6). This is consistent with our FCS data shown in Fig. S5b and the state diagram shown in Fig. S5a. We have now included a statement in the main-text that reads:

“*We independently confirmed that poly(rU) RNA does not significantly interact with the PLP by using fluorescence correlation spectroscopy (FCS), which revealed identical PLP autocorrelation curves in the absence and presence of poly(rU) RNA (Fig. S5b) in the single phase regime. Furthermore, partition analysis of an RNA oligomer (rU10) showed a partition coefficient of ≤ 1.0 in PLP droplets (Fig. S6). Combining the state diagram, FCS, and partition analyses, we conclude that poly(rU) RNA does not have significant interactions with the PLP.*”

9. I have serious problems with referring to condensates as being micellar. First, micelles are known as pseudo-phases. Second, micelles will have a fixed size and this size as well as the molecularity of the micelle will be governed by molecular dimensions of the components. There is no way a micellar system can make micron-scale condensates.

Third, do the sizes of condensates increase with concentration or do the sizes stay fixed and the numbers of condensates increase? If this is the case, these are micellar condensates and the phase separation is an exemplar of micro-phase separation. The Chilkoti lab has shown how this can be realized in block co-polypeptide systems. Fourth, if the simulations are suggestive of micellar architectures, then the question is these are indeed *bona fide* micelles and if yes, how does one justify micelles growing to be spherical condensates and not forming cylinders, toroids, or bicontinuous phases? It is more than likely that the observed micellar features are largely a consequence of the serious finite size artefacts that prevail in the simulations.

We agree with the reviewer's statements regarding micelles. However, we used the term "micellar condensate" to describe condensates that share some characteristics with micelles but are formed by liquid-liquid phase separation and are not micelles in a strict sense. Micelles are indeed small; they do not grow in an aqueous solvent; their surfaces are enriched with more hydrophilic segments of the polymers while their cores are enriched with more hydrophobic segments. Upon increasing the total polymer concentration, the number of micelles increases while their sizes remain the same, as described in the seminal works by Israelachvili and Tanford (Israelachvili et al., *Journal of the Chemical Society, Faraday Transactions 2: Molecular and Chemical Physics* 72 (1976): 1525-1568 and C. Tanford, *J. Phys. Chem.*, 1974, 78,2469). According to our proposed model, which is inspired by the theoretical work of Shklovskii and Zhang (Zhang, R., & Shklovskii, B. I. (2005). *Physica A: Statistical Mechanics and its Applications*, 352(1), 216-238), polyelectrolyte condensates prepared at disproportionate mixture composition show spatial (i.e., core-shell) organization where the core is enriched with fully screened complexes of proteins and RNA (net formal charge is zero; lower solvation volume). The surface is enriched in partially complexed chains (non-zero net formal charge; higher solvation volume). This structural organization is also consistent with a recent work by Harmon, Holehouse, and Pappu (Harmon, T. S., Holehouse, A. S., & Pappu, R. V. (2018). *New Journal of Physics*, 20(4), 045002.). In three preceding papers, we have provided experimental evidence that oppositely charged proteins and RNAs without co-block architecture can form droplets that show arrested growth due to enhanced repulsion between like charges that accumulate at the droplet surface at disproportionate mixture composition (Banerjee et al., 2017, *Angew Chem Int Ed Engl.* 56(38):11354-11359; Alshareedah et al. 2019, *J. Am. Chem. Soc.* 141, 37, 14593-14602; Alshareedah et al. 2020, *Proc. Natl. Acad. Sci. (U.S.A.)*. 117 (27) 15650-15658). Therefore, to inform the readers about the common characteristics between these condensates and micelles while preserving the fact that these are condensates formed by liquid-liquid phase separation, we chose the term "micellar" condensates; but we agree that there is a clear distinction between calling these structure "micelles" and calling them "micellar condensates".

Given the focus of our current manuscript is not on the micellar condensates but the peptide-RNA condensates displaying a core-shell structure at disproportionate mixture composition, we have eliminated the term "micellar condensates" and replaced it with "spatially-organized condensates". We hope with these additional clarifications and revisions; we have addressed the reviewer's concern.

10. “To validate the micellar condensate formation in our system, we turn to molecular dynamics simulations...”. This phrase is problematic on so many levels. First, see the concerns raised above. Second, one should never use the term validate. It conveys the impression of certitude. The better alternative is to state that “in order to provide a molecular-level understanding of...”

We agree with the reviewer that the alternative wording is a more accurate reflection of the way we have used simulations in this work that is to provide *a molecular-level understanding* and to dissect which forces are important for reproducing the observed multi-phase condensate topologies in computer simulations. The revised statement now reads:

“In order to provide a molecular-level understanding of these assemblies, we performed molecular dynamics (MD) simulations using...”

11. The cartoon representations in Figures 1e, 4e, and 5d seem to convey the impression of essentially vesicular structures. Almost certainly there will be molecular scale or larger scale roughness on the surface of the condensates, and this would be indicative of a mixture (favoring one species over the other) of compositions at the surface. If that is not the case, then the interfacial tensions would be rather high, and all measurements to date place these numbers at being 6-7 orders of magnitude smaller than that of the air-water interface. This is important because a casual reader will walk away with the wrong impression.

We have now revised the cartoon representations in Figures 1e, 4e, and 5d to address the concern raised by the reviewer here.

12. Figure 1g is important, and in the context of what is now known, these data suggest that condensate formation, at least under the conditions investigated, is probably under the exclusive control of heterotypic interactions in the RLD-RNA system. If this were not the case, one would not observe a one-to-one correspondence between the bulk ratios of RNA to RLD on the partitioning of PLDs into these condensates. This point has to be emphasized more thoroughly.

In the revised manuscript, we have now emphasized the point that the ternary system phase behavior and compositional control is governed primarily through the heterotypic interactions in the RRP-RNA system. The revised statement now reads

“...As such, the phase behavior and compositional control of the PLP-RRP-RNA ternary system are expected to be governed primarily through heterotypic interactions between RRP and RNA. To test this idea, we next examined the impact of RNA on the phase behavior and organization of PLP-RRP condensates.”

13. The authors, based on simulations in systems that are likely plagued by finite size issues, propose, essentially assert that the sites on the surfaces of RLD-RNA condensates are the relevant species. Why should this be true? Figure 1j would appear to offer experimental data in support of the contention, but the level of surface enrichment is a) rather modest, b) system-specific, and c) does not account for the relative differences in the number of sites at the interface vs. the interior, which would scale as $R^{2/3}$ where R is the radius of the condensate. And the system specificity might arise due to comparisons being made at nonequivalent quench depths and / or not normalizing the effects in terms of the numbers / concentrations of aromatic stickers. This raises serious questions about Figure 1k as well and I fear that the surface-centric narrative is being pushed largely by the features identified in simulations plagued by finite size considerations. In fact, pure surface binding cannot enable the condensate switching observed in this work.

The experimental results that we reported throughout in Fig 1 and Fig 4 in the main-text provide experimental evidence that RRP-RNA condensates have spatially-organized structures where the surface of these condensates are occupied by partially condensed RRP (at low RNA) or RNA (at high RNA) chains. This proposition is derived from the earlier works of Shklovskii and Zhang (Zhang, R., & Shklovskii, B. I. (2005). *Physica A: Statistical Mechanics and its Applications*, 352(1), 216-238), and asserted by the work of Harmon et al. (Harmon, T. S., Holehouse, A. S., & Pappu, R. V. (2018). *New Journal of Physics*, 20(4), 045002), where it was shown that if a condensate is hosting two populations of charged and uncharged species, the charged species would preferentially accumulate near the interface where the solvent exposure is maximized. This is driven by differential solvation as argued elegantly by Harmon and colleagues. Along this line, in our recent work on protein-RNA hollow condensates (Alshareedah et al. 2020, *Proc. Natl. Acad. Sci. (U.S.A.)* 117 (27) 15650-15658), we argued for a similar phenomenon where partially condensed complexes are preferentially located near the surface of the condensates since they have a non-zero net charge and surface localization enables a higher solvation volume. This was done in an effort to explain the arrested growth of condensates formed at a highly disproportionate mixture composition as well as the formation of hollow vesicle-like condensates (Alshareedah et al., *PNAS* 2020). In the current work, we provided further experimental evidence, by devising a PLP recruitment strategy in RRP-RNA condensates at a variable RNA-to-RRP ratio. Given PLP preferentially binds to RRP but not RNA, we see a surface enrichment of PLP chains into RRP-rich RRP-RNA condensates at PLP concentrations ($\leq 1.0 \mu\text{M}$) that are much lower than PLP saturation concentration ($>100 \mu\text{M}$). This surface recruitment is absent at high RNA conditions, lending clear support to our proposed structural model of RRP-RNA condensates which suggests that the condensate surface at high RNA is enriched with RNA, while at high RRP, the surface is enriched with RRP. The lack of PLP enrichment in RRP-RNA condensates is readily explained by the insignificance of RNA-PLP interactions, the experimental evidence for which is shown in state diagrams and also using FCS and partition analysis (Figs. S5&S6). Finally, we note that a similar mechanism of surface enrichment into a protein-RNA condensate was recently reported by Regy and colleagues using MD simulation (Regy et al., (2020), *Nucleic Acids Research*, gkaa1099, <https://doi.org/10.1093/nar/gkaa1099>), lending further support to our proposed model.

Besides, the reviewer has three points of concern on these results, which we discuss point-by-point below:

- (a) “the surface enrichment is rather modest”: We see that the fluorescent intensity of Alexa488-labeled PLP is higher on the surface of the RRP-rich condensates than in the core. The difference is about a 4-fold for the case of EWS^{PLP} and ~ 2 fold for the case of BRG1^{LCD} and FUS^{PLP}. This difference was observed over the entire sample and the ratio of PLP intensity at the surface and the core was quantified for at least 70 condensates (Fig. S14). Accordingly, we believe that this difference, although small in magnitude, is statistically significant and should be addressed. We also note that our results are unique, but not unexpected and bear similarity to the recent findings reported by Parker and Colleagues (Tauber et al., 2020, Cell 180, 411–426; <https://doi.org/10.1016/j.cell.2019.12.031>). In that work, the authors showed enhanced surface localization of several fluorescently labeled mRNAs to RNA-only condensates *in vitro* (formed by poly(A) RNA) as well as RNP condensates such as purified stress granules from mammalian cells. We have stated this clearly in the revised manuscript and included this citation. Furthermore, we have added new experimental data showing that this surface enrichment of PLP also occurs in RRP-poly(phosphate) condensates (Fig. S15) prepared at low poly(phosphate) level but not at high poly(phosphate) level. The results that differential surface vs. core localization of the PLPs are also observed for condensates formed by a linear polyanion that lacks RNA bases lends further support to our proposed model.
- (b) “the surface enrichment is system-specific”: The condensates, shown in Figure-1g-i and analyzed in Figure-1j&k, are prepared at identical concentrations (the same quench depth within the RRP-RNA state diagram). These are RRP-RNA condensates at low and high RNA-to-RRP ratios. The PLPs were used at concentrations that are orders of magnitude lower than the PLP saturation concentration (we have studied 3 PLPs and a 4th π -rich system consisting of aromatic stickers: RNA Pol II CTD). In all the cases, we see a significant difference (2-4 fold) in the PLP enrichment at the surface and the PLP enrichment in the core of RRP-RNA condensates. The changes in the magnitude of this difference (i.e., surface vs. core enrichment) as we change the PLP identity can be attributed to differences in the interaction strength between RRP and the various PLPs used in our study. Such a system-specific behavior is not unexpected and was also noted in Tauber et al. for mRNA partitioning on the surface vs. core of RNA condensates (2020, Cell 180, 411–426). Importantly, the observation that there is a difference between the PLP localization to the surface vs. the core of RRP-RNA condensates holds for all the tested PLP chains with aromatic stickers in our study as well as RRP-polyphosphate condensates. Accordingly, we argue that the magnitude of difference between the enrichment at the condensate surface vs. condensate core may be system-specific, but the existence of this difference is general and can be explained by our proposed model of spatially-organized RRP-RNA condensates.

- (c) “the surface enrichment does not account for the relative differences in the number of sites at the interface vs. the interior, which would scale as $R^{2/3}$ where R is the radius of the condensate”: To our understanding, the reviewer is indicating that the number of binding sites at the surface is different than the number of binding sites at the core and that the difference in the number of binding sites between the surface and the core changes with the condensate size (since for a sphere, $surface\ area \propto V^{2/3}$). This geometric consideration is correct. Accordingly, if there is no difference in the composition between the surface and the core of an RRP-RNA condensate (i.e., if the condensate is structurally isotropic instead of spatially-organized), the intensity coming from the partitioning of PLP in those condensates should be higher in the core and lower at the interface. In our experiments, we see a recurring trend: the intensity near the surface is ~ 2-4 fold higher than the intensity near the core. Our proposed model suggests that RRP-RNA condensates assume spatially-organized core-shell architecture. Therefore, the most plausible explanation for the surface localization of PLP chains is manifested due to an increase in the number of *available* PLP binding sites (# of Arg-residues) at the surface, especially at low RNA-to-RRP ratios. On the other hand, the arginine residues at the core are engaged in RNA binding, and hence, are not available to PLP chains. Hence, PLP would not partition strongly in the condensate core due to the competition with RNA. This scenario is further corroborated by a complete lack of surface enrichment of PLPs at high RNA conditions, where the surface is expected to be enriched in free RNA segments that do not interact strongly with PLP chains. Lastly, we reiterate that we do not claim that the surface enrichment for all PLPs is identical, there is system specificity that affects the magnitude of the relative surface vs. core enrichment (which may also depend on the size of the condensates as the reviewer rightly suggested), but the existence of a disparity in PLP localization to the surface and the core seems to be true for all the tested PLPs as well as RRP-poly(phosphate) condensates.

By the same mechanism, the organization of the multiphasic PLP and RRP-RNA condensates responds to the RNA level due to the changes in RRP-RNA surface composition (Fig. 4). These experimental results, as the Reviewer pointed out in point #20 below, stand without the need for MD simulation. We propose that such changes in surface composition of RRP-RNA condensates lead to changes in interfacial tension as a function of RNA-to-RRP ratio, which can drive a clear morphological transition (Fig. 4b&c) as our fluid-interface simulations suggest (Fig. 4d).

To address the reviewer's point, we have provided a discussion of system specificity in these experiments in the relevant sections of the manuscript. The revised statement now reads: *“Although the magnitude of relative surface enrichment of PLP and π -rich clients seem to vary with the client used (Fig. S14), the existence of such surface enrichment is general to the tested client proteins. Such system specificity may arise from the varying interaction strength between RRP and the different client proteins. We further confirm that this surface enrichment is not specific to RNA by repeating the same assay with RRP-poly(phosphate) condensates (Fig. S15). We note that our results of PLP client recruitment preferentially on the surface of RRP-RNA condensates bears similarity to a recent report of enhanced surface localization of several fluorescently labeled mRNAs to RNA-only condensates in vitro formed by poly(rA) RNA as well*

as RNP condensates such as purified stress granules from mammalian cells⁶². Similar surface localization was also observed in MD simulations of condensates formed by Arg-rich disordered proteins and polynucleotides⁶³.”

Regarding finite-size issues of MD simulations, please see our response to **point 20 below**.

14. The effect of condensate dissolution by RNA, for which the authors have unique data sets, should be analyzed using the polyphasic linkage formalism. The protocol prescribed by Ruff et al., would be helpful as would the work of Posey et al. (see: <https://www.jbc.org/content/293/10/3734>).

We thank the reviewer for this suggestion. Our current manuscript attempts to provide a molecular-level picture of how sequence-encoded and composition-dependent protein-RNA interactions control the multiphasic architecture of condensates. We have now performed qualitative analysis using polyphasic linkage formalism which suggests that RRP acts primarily as a ligand that binds to the PLP preferentially in the dense phase. This is consistent with the fact that we do not observe the formation of homotypic condensates by the R-rich polypeptide in our experimentally-probed concentration regime.

We have cited these two relevant papers and revised our discussion on the results, which now reads:

*“Our analysis revealed that the phase separation of PLP is enhanced with RRP in a composition-dependent manner. Specifically, the LLPS concentration threshold for PLP (C_{LLPS}^{PLP}) decreases monotonically with increasing RRP concentration [in the absence of the RRP, $C_{LLPS}^{PLP} = C_{saturation}^{PLP} \sim 240 \mu\text{M}$; at an RRP-to-PLP ratio of 5, $C_{LLPS}^{PLP} \sim 120 \mu\text{M}$] (**Fig. 1a**; **Figs. S1&2**) under our experimentally tested conditions (up to RRP-to-PLP ratio of 20:1). Although PLP readily undergoes homotypic condensation in the absence of RRP, the latter did not show any sign of homotypic LLPS at the concentrations used in our study. Hence, our results that the saturation concentration of PLP decreases monotonically with [RRP] are consistent with a scenario where RRP acts as a ligand that binds to the PLP preferentially in the dense phase^{52,53}.”*

We also included a plot in the SI (**Fig S2b**) showing how C_{LLPS}^{PLP} is varied as a function of [RRP]. Additionally, we changed the units in a way that reflects the concentration of “stickers” [i.e., the concentration of Tyr (primary stickers) in PLP and concentration of Arg in RRP] in our system. A more quantitative analysis of the ternary phase behavior of our PLP-RRP-RNA system using polyphasic formalism is beyond the scope of the current manuscript and would be a better fit for a future study.

15. The term client is a misnomer given the observations of preferential binding, dissolution, and sorting. By the definitions of Rosen and colleagues, a client is simply a molecule that has a PC > 1. What the authors are reporting is really a ligand-aided sorting mechanism via polyphasic linkage followed by contact without wetting of unmixed condensates. This behavior has been reported by the Gladfelter lab in a series of reports dating back to 2013. These papers have been ignored in the narrative, and this error of omission has to be remedied.

We thank the reviewer for raising this point. Indeed, we followed the common definition of a client, according to which client is a molecule that is not essential to the formation of the homotypic PLP and heterotypic RLP-poly(rU) condensates (such as Pol II CTD and/or short ssRNA) but they are preferentially included within the condensate ($PC > 1$). This definition is equivalent to the definition proposed by Rosen et al. and by Holehouse and Pappu (Banani et al., 2016, *Cell* 166, 651–663, Holehouse and Pappu *Biochemistry* 2018, 57, 17, 2415–2423). Hence, through RNA-induced demixing of PLP and RRP, we are sorting the biomolecules (Pol II CTD and ssRNA), which are not essential to the formation of PLP-RRP condensates but preferentially included within PLP-RRP condensates, into two coexisting condensates. Hence, the term “client” was used. In the revised manuscript, we revised the term “client sorting” and replaced it with “biomolecule sorting”.

We have also cited two relevant papers by the Gladfelter lab (Langdon, E. M. et al. (2018). *Science*, 360(6391), 922-927 and Gasior, K. et al. (2020). *Bulletin of Mathematical Biology*, 82(12), 1-16). We hope this will address the reviewer’s concern.

16. When the authors introduce the nomenclature for comparing interfacial tensions, it is unclear if they are referring to a binary mixture. This is what the A, B, and C designations appear to suggest. If so, the relevance of the inequalities is lost on me for a ternary system. Also, the authors do not once mention Neuman’s inequality nor do they connect with the copious literature on this subject.

We thank the reviewer for pointing out this problem. What we meant by A, B and C are three phases, not three components. We have four components (solvent, PLP, RRP, RNA) that undergo phase separation into three coexisting phases (dilute phase, RRP-RNA condensed phase, and PLP condensed phase). However, the reviewer is right that this nomenclature might be confusing, and therefore, we added a sentence in the revised manuscript describing the nomenclature in detail. We now used four letters, A, B, C, D to describe the three phases A, B+C, and D. In our original submission, we did not go into detail about how this relation (Neumann’s inequality) is obtained, and simply cited Guzowski et al. (Guzowski et al. *Soft Matter* 8, 7269-7278 (2012)) and a seminal book by Rowlinson and Widom on this topic (Rowlinson, J. S. & Widom, B. *Molecular theory of capillarity*). Both of these references have an extensive discussion on the Neumann triangle and the origin of the aforementioned equation. In the revised manuscript, we have included another seminal work as a citation in this context (Lester, G. *Journal of Colloid Science* 16, 315-326 (1961)). We note that the discussion of contact angles and surface tension is not related to the number of components but the number of phases, or the number of liquids at equilibrium. We asserted this distinction in the revised manuscript, included a discussion on the Neumann triangle, and pointed the readers more clearly to the abovementioned references.

The revised text now reads:

“For a four-component system (A, B, C, D) that undergoes phase separation into three phases- A (condensate), B+C (condensate), and D (dispersed liquid phase), the equilibrium morphology is determined by the relative interfacial tensions (γ_{A-D} , γ_{B+C-D} , and $\gamma_{A-(B+C)}$) of the three liquid phases^{11,15,16}..... The contact angles are the dihedral angles formed by the liquids at the

three-phase boundary. Since the forces acting on the three-phase boundary should sum to zero (Neumann triangle), relations between the contact angles and the interfacial tensions can be derived^{64,68,69}. Therefore, the cosine of a contact angle (θ), can be expressed as a function of the interfacial tensions of the three liquid phases as^{64,68}”

17. The schematic in Figure 4a seems to be straight out the review published by Shin and Brangwynne. If the authors wish to use this graphic, then they should do so with attribution, the permission of the authors, and permission from Science magazine. Otherwise, they should revise the graphic and still cite the Shin and Brangwynne paper in the figure caption.

We agree that Figure 4a in our manuscript resembles the schematic in the review article by Shin and Brangwynne, which was not our intention. By no means, it was taken straight out of the Shin and Brangwynne paper. In our scheme, we choose the colors of two condensates to be *red* and *green* because those are the colors of our experimentally imaged condensates by confocal microscopy. Coincidentally, the same color scheme was used by Shin and Brangwynne, which might be the source of confusion. We have now revised this figure to make it as distinct as possible, which should address the reviewer’s concern here.

18. Equation (1) introduces three interfacial tensions viz., γ_{RLD} , γ_{PLD} , and $\gamma_{PLD-RLD}$. The narrative would suggest that γ_{PLD} is the interfacial tension that describe the interface formed between PLD condensates and a dispersed phase that is solvent-rich and PLD-deficient. Likewise, $\gamma_{PLD-RLD}$ should be the interfacial tension at the interface between PLD-RLD condensates and a dilute solution comprising PLD and RLD molecules. So, what then is γ_{RLD} ? Afterall, the RLD does not form condensates with a defined phase boundary. So, this equation is probably incorrect. This impacts all of the analysis presented in Figure 4 and it should be clarified and revisited.

We apologize for the confusion here; γ_{RRP} should be $\gamma_{RRP+RNA}$. We assumed that it is clear that we are referring to RRP+RNA condensates as RRP condensates given RRP does not form homotypic condensates within the concentrations and buffer conditions utilized in our work. Therefore, γ_{RRP} indicates the interfacial tension between RRP-RNA condensates and the solvent, and $\gamma_{PLP-RRP}$ indicates the interface between PLP condensates and RRP-RNA condensates. We have now corrected this throughout the text. We used $\gamma_{RLD+RNA}$ to represent the interfacial tension between RRP-RNA condensates and the solvent, and $\gamma_{[RRP+RNA]-PLP}$ to represent the interface between RRP-RNA condensates and PLP condensates. The equation is written in the revised manuscript as

$$\cos(\theta_{PLP}) = \frac{\gamma_{RRP+RNA}^2 - \gamma_{PLP}^2 - \gamma_{PLP-[RRP+RNA]}^2}{2 \gamma_{PLP} \gamma_{PLP-[RRP+RNA]}}$$

19. With regard to the differences between Lys and Arg, the recent work of Greig et al., is very much in the spirit of the findings reported here, and they have been ignored. This should be remedied. Please see: <https://doi.org/10.1016/j.molcel.2020.01.025>. The authors are

advancing the importance of sequence-encoded interactions as determinants of condensate regulation, morphology, and spatial organization. This points to cellular control over the molecular grammar and accordingly, it is important to ensure that preceding contributions not be ignored or given short shrift. Apportioning due credit does not take away from the current work, it instead buttresses its importance.

The Arg-to-Lys mutation was used in this study from a design perspective, to weaken the interactions between RRP and RNA. In this regard, the reviewer is right that the work of Greig et al. is an important advancement to the field of biomolecular condensates. We thank the reviewer for drawing our attention to this error and we have cited it in the revised manuscript.

20. With all due respect, the data stand on their own without the MD simulations. To assert otherwise is misleading. The key insight that emerges from the simulations is surface centric view, and this should, at this juncture be framed more as a hypothesis rather than fact because this is where the data are particularly unclear. If the authors wish to assert otherwise, then it would help immensely to perform computational interrogations that lead to predictions of the relative interfacial tensions for different condensates whilst taking care to guard against finite size artefacts. This is a tall order, and in my view, it is beyond the scope of the current work. Instead, I propose toning down the pronouncements regarding the predictive utility of the MD simulations. These simulations provide a picture that may or may not be true and their accuracy remains undetermined at this time.

We agree on the fact that simulations using this system size are not compatible with mesoscopic scales used in experiments and are likely to have finite size issues when probed for detailed chain packing on the surface. That being said, however, these simulations do indeed capture the formation of interfaces and multi-phase topologies which are remarkably consistent with the experimental observations. This insight is significant as it reveals sufficient resolution and sequence specificity for forming the observed multiphase topologies. Thus while we have used finite size systems, we still managed to obtain the relevant coarser observables for this study such as the trends of multiphase droplet topology as a function of RNA and sequence mutations targeting cation- π interactions. The simulations use coarse-grained models of IDP and RNA chains hence the power of their generality implying that many systems with similar charge patterning and chain length and elasticity will very likely display the same condensation topologies.

Given that MD simulations provide models for thinking and it is ultimately the experiment upon which these results are meaningful, we have therefore revised the text to stress this relationship between experiment and simulation in the paper. We have stressed that in our study, the MD simulations were used to attain an extra insight into the molecular driving forces in ways which can only be obtained through simulations: that is using Occam's razor to peel the molecular complexity distilling the observed experimental condensate topologies to minimal physical driving forces. For instance, comparing generic mean-field and cation- π enhanced potentials involving LYS vs ARG shows that only the latter is capable of reproducing the correct pattern of multiphase topologies. Hence these findings shed some light on the key forces at play.

21. The authors propose that fuzzy interactions, non-stoichiometric interactions are the central determinants of the numerous interesting observations they present in this work. What makes interactions fuzzy and how does one quantify such fuzzy interactions? It would have to be through the device of distribution functions. If so, then are the effects of interactions quantified using pair distribution functions in liquids to be referred to as being fuzzy? I shouldn't think so and this nomenclature is not helpful because we end up with a conflation of thermodynamics and dynamics concepts that are not easy to follow. Additionally, the assertion regarding interactions being non-stoichiometric is a non-quantitative one. As has been noted recently, the numbers (valence) of stickers and the modulation of their interactions by spacers directly influence the driving forces for phase separation aided percolation transitions. These interactions can give rise to stoichiometric assemblies or non-stoichiometric assemblies that are highly networked. This would imply that it is misleading to refer to interactions as being non-stoichiometric. If the authors wish to use this verbiage, then they should explain exactly what they mean by this usage.

We agree with the reviewer that the term “fuzzy interactions” and the non-stoichiometric nature of interactions were not explained properly in our original submission. The concept of “fuzzy interactions” between intrinsically disordered RRP and RNA chains reflects our assumption that the resulting polypeptide-RNA complexes exhibit a structural continuum. This assumption was inspired by the work of Monica Fuxreiter and Peter Tompa (there are several papers, for example, see *Trends Biochem. Sci.*, 33 (2008), pp.2-8 and *J Mol Biol* (2018) DOI: 10.1016/j.jmb.2018.02.015). The fact that RRP and RNA mixtures form liquid droplets and hollow vesicle-like condensates at disproportionate mixture compositions (Alshareedah et al, 2020, PNAS), we proposed that the complexes formed by the disordered R-rich polypeptide and RNA chains in our system are not fixed but vary continuously with mixture composition, signifying the existence of a structural and/or dynamical continuum in this ensemble. While alternative explanations may exist to explain our results, we think the proposal of “fuzzy complexes” by Fuxreiter and Tompa sheds light on our experimental observations. Furthermore, theoretical work by Muthukumar and colleagues [Adhikari et al. *J. Chem. Phys.* 149, 163308 (2018) and Muthukumar, *Macromolecules* (2017), 50, 24, 9528–9560] on polyelectrolyte complexes have invoked a similar idea of an ensemble of partial complexes along with Shkolvskii and Zhang ((2005). *Physica A: Statistical Mechanics and its Applications*, 352(1), 216-238). We note that a statistical mechanical description of the complete ensemble of such complexes does not exist yet, hence the use of the term “fuzzy complexes” is rather qualitative. On the other hand, the term “non-stoichiometric interaction” was used to describe the formation of non-stoichiometric assemblies.

In the revised manuscript, we clarified the term “fuzzy interactions”, cited the relevant literature, and omitted such terminologies where it is deemed unnecessary. The revised text now reads:

“However, our results presented here for RNA-RRP condensates’ multiphasic patterning with PLP condensates reveal a unique role of the condensates’ surface organization, which can be manipulated by varying the mixture composition. This is likely to be a direct result of the existence of a structural continuum⁸¹ in the ensemble of RRP-RNA complexes where the

stoichiometry of the resulting complexes is sensitively dependent on the mixture composition^{28,54,82,83}.

Semantic and conceptual issues

1. There is the repeating usage of the phrase “networks of overlapping interactions”. This is confusing. What does the term “overlapping” mean in this context? Please clarify and / or rephrase. Close reading suggests that the authors might be referring to concentration (or stoichiometry) dependent competition between competing interactions (maybe).

We have revised “overlapping interactions” to “competing interactions”.

2. Please use the more widely accepted term biomolecular condensates instead of biocondensates. And, given that BMC is an abbreviation that has been co-opted for other topics in biology, it is best to adopt the current working approach and use the term condensates if one wants to avoid using the term biomolecular condensates due to space considerations. Please do not use the BMC abbreviation. This makes things confusing.

We have revised our manuscript and omitted the abbreviation BMC. Instead, we used biomolecular condensates (or simply “condensates”)

3. The abbreviation RLD is also non-standard and it does not fit as an abbreviation either for R-rich polypeptide or for R-rich low complex domain. The better approach would be something like RRD or RRP for R-rich domain or R-rich polypeptide.

We have chosen the term “RRP” for R-rich polypeptide and “PLP” for Prion-like polypeptide. We also used KRP for K-rich peptides.

4. Does the term “topology” best describe the mesoscale organization or structure of condensates?

We have replaced the term “topology” with “morphology”.

5. Please replace liquid-liquid phase transition with one of the following: (i) phase transition or (ii) liquid-liquid phase separation or (iii) phase separation aided percolation.

Following the reviewer’s suggestion, we have replaced “liquid-liquid phase transition” with liquid-liquid phase separation.

6. In motivating one’s work, it is common practice to set things up with the typical declaration that something is “largely unexplored” or “poorly understood” or “controversial” etc. The authors take a similar approach and it does disservice to what has come before and the very real possibility that the current work builds on what has come before. Therefore,

simply state the phenomenon one wishes to explain or describe and state something like “Here, we use a tractable, minimalist ternary system to ...” – as in state what the paper is about without anointing it as being novel or poorly understood or hitherto unexplored. If one broadens one’s horizons, it becomes patently obvious that the concepts have been very well established in the context of synthetic systems that undergo complex coacervation. The work stands on its own. No need for unnecessary edification.

The reviewer is right. Although it is a common practice to use phrases such as “poorly understood” to set things up in manuscripts, we agree that it is unnecessary and should be avoided. It is best to let the data speak. Ultimately, any work should stand on its own merit, as noted by the reviewer in the context of our current manuscript. We have addressed this concern in the revised manuscript and eliminated the statement of concern.

7. The authors list “ π rich” electron systems Y/P/Q/G/S as being enriched in LCDs. If the authors are referring to PLDs, then Pro definitely is not prominent in these sequences. Perhaps the authors mean F for Phe. Also, Asn shows up more frequently in *bona fide* PLDs than Gln. Please see the work of Alberti et al., published in Cell and Halfmann et al., published in Molecular Cell over a decade ago. Finally, it is not clear that Q, G, and S are best described as being “ π rich” systems. This nomenclature has been suggested by Vernon et al., but it this is disputable. From a zeroth order perspective, it would be best to refer to these residues as polar residues capable of forming hydrogen bonds.

We have revised this description here and cited the two papers mentioned by the reviewer. The revised statement now reads “*PLPs are typically characterized by the presence of π electron-rich and polar amino acids^{33,34} (Y/N/Q/G/S; examples: hnRNPA1, TDP43, FUS)²³*”

8. In what sense are R-rich LCDs (please change this phrase) “promiscuous RNA binders”?

The term “promiscuous RNA binders” was used based on previous literature reports on the RNA binding properties of RGG-box of RNPs (Ozdilek and Schwartz, Nucleic Acids Res. 2017 Jul 27; 45(13): 7984–7996). We have revised it to “*whereas R-rich polypeptides bind RNAs with a broad range of sequence composition and structures³⁵, and commonly occur as intrinsically disordered RGG domains^{22,36} (examples: G3BP1, LSM14A, hnRNPD, EWSR1, FUS, TAF15).*”

9. The term “promiscuous” is thrown around somewhat loosely. In this context, it is misleading, and it often leads to the trope of IDR-mediated interactions as being nonspecific. In the context of the current work, the authors wish to imply that there are likely to be competing interaction networks and that they intend to investigate the consequences of these competing effects.

The term “promiscuous” was used in two places in our manuscript. However, we acknowledge the point raised by the reviewer here and omit the use of promiscuous non-specifically in our manuscript.

10. The italicized phrases “monotonically” and “highly non-monotonic” should be non-italicized and are actually misleading because the it is a tautology that if the “RLD” is in vast excess when compared to the PLD, then even the PLD-RLD system will show reentrant behavior or the RLD will end up wetting the PLD. The concentration regimes where this might be realized have not been explored in the current work and therefore one should be cautious about making these assertions.

Even at 20-fold excess RRP, we have not seen a reentrant behavior for the PLD-RRP system. However, we acknowledge that one might see such behavior at even higher RRP concentrations. To make this clear, we have revised our statement which now reads:

“the LLPS concentration threshold for PLD (C_{LLPS}^{PLP}) decreases monotonically with increasing RRP concentration [in the absence of the RRP, $C_{LLPS}^{PLP} = C_{saturation}^{PLP} \sim 240 \mu M$; at an RRP-to-PLP ratio of 5, $C_{LLPS}^{PLD} \sim 120 \mu M$] (Fig. 1a; Figs. S1&2) under our experimentally tested conditions (up to RRP-to-PLP ratio of 20:1).” We have also non-italicized phrases “monotonic” and “non-monotonic”.

11. The word uptaken should be taken up and there is one place where the authors refer to deferential partitioning, which should be differential partitioning.

We have revised this statement to address this issue.

12. In many of the figure captions, the authors write that “each type of droplets was...”. This should be fixed to “each type of droplet was...”.

We have revised this statement to address this issue as suggested by the reviewer.

Reviewer #2 (Remarks to the Author):

This is one of the best studies I read recently. This work adds significantly to the field and will have a noticeable impact. It is nicely written and contains an impressive amount of convincing data. The presented statistical analysis is appropriate and valid. The level of details provided in the manuscript is sufficient to reproduce the work. This work is novel and convincing, and data described will be of interest to others in the community and the wider field.

We thank the reviewer for his/her kind words, constructive comments and suggestions. We are glad to learn that the reviewer has found this work important for a wider community.

Reviewer #3 (Remarks to the Author):

There is currently great interest in understanding molecular mechanisms of liquid-liquid phase separation of protein/RNA mixtures. In this manuscript, Kaur et al. have examined a well-defined ternary system consisting of a low-complexity (prion-like) polypeptide (PLD), Arg-rich Polypeptide (RLD) and RNA. Analysis of experimental data, together with the results of MD simulations, allowed the authors to explain, in molecular terms, the topology of these three-component condensates and the mechanisms that control this topology. Overall, the manuscript is very interesting and clearly written, and the main conclusions well supported by experimental data. This work represents a significant advance in our understanding of multi-component condensates.

We thank the reviewer for his/her kind words, constructive comments, and suggestions.

Specific points:

1. My only significant concern is the lack of a more comprehensive discussion of the present findings in a biological context. Specifically, proteins driving liquid-liquid phase separation in cells typically contain both low-complexity and RNA-binding domains. It would be helpful if the authors could discuss how their findings using the isolated domains (or models thereof) alone could explain the behavior of systems containing full-length proteins.

We thank the reviewer for this suggestion. Indeed, our results show that multiphasic behavior can be established using only disordered domains of RNPs and model homopolymeric RNAs. We speculate that structured domains would add more specificity to the multiphasic behavior and also to the material nature of cellular condensates. This idea is based on the elegant work of Feric et al. (Feric, M. et al., (2016), *Cell*, 165(7), 1686-1697) where they show that for the N-terminal Arg/Gly-rich disordered domain of FIB1 nucleolar protein, the condensates showed purely viscous behavior. Adding the C-terminal methyltransferase domain resulted in a significant enhancement of viscoelasticity in FIB1 condensates. The authors went further to show that FIB1 (fibrillarin) is essential for the nucleolar structure as it is the major component of the dense fibrillar core, however, the structured domain is not essential for the multiphasic structure of the nucleolus as their deletion constructs indicate. On the other hand, NPM1 (nucleophosmin), the major constituent of the granular component, preserved the multiphasic nucleolar structure only in the presence of its oligomerization domain and its RNA recognition motifs (RRMs), both of which are structural domains. Therefore, we believe that the role of structured domains is related to their function, and fine-tuning the composition and organization of multiphasic condensates. Based on the reviewer's recommendation, we have now included a short discussion in the conclusion section of the manuscript regarding the effect of structured domains. The revised text now reads: *"Lastly, LLPS-driving proteins often feature modular architecture with both low-complexity disordered domains and structured modules. The presence of these structured domains (such as RNA recognition motifs) is expected to alter the multiphasic behavior and condensate properties, especially when these domains are involved in the interactions stabilizing the condensates."*^{11,83}

Minor points:

1. Fig 1: From the microscopic images shown in panels g-i it appears that the proportion of protein that accumulates within the droplets' interior is substantially larger for BRG1 LCD than for two other proteins studied. Some explanation of this quantitative differences should be provided.

We thank the reviewer for this remark. We propose that these differences are a manifestation of the following factors: (a) The sequence-dependent interactions between the arginine-rich polypeptide (RRP) and the Prion-like polypeptide (PLP) chains; for example, the number and frequency of tyrosine residues in FUS^{PLP} and EWS^{PLP} are greater than the BRG1^{LCD}, which is enriched in proline residues. (b) The size of the droplets; the larger the droplet is, the greater is the disparity between the core and the interface. Larger droplets host a larger number of partially complexed protein-RNA chains on the surface and hence the difference in density of available binding sites near the interface may increase with increasing the size of the condensate. To explain this, we have now added a short discussion on the difference in surface enrichments across the different PLPs.

The revised text now reads: *“Although the magnitude of the relative surface enrichment of the PLP and π -rich clients seem to vary with the client used (Fig. S14), the existence of such surface enrichment is general to the tested client proteins. Such system specificity may arise from the varying interaction strength between RRP and the different client proteins..... We note that our results of PLP client recruitment preferentially on the surface of RRP-RNA condensates bears similarity to a recent report of enhanced surface localization of several fluorescently labeled mRNAs to RNA-only condensates in vitro formed by poly(rA) RNA as well as RNP condensates such as purified stress granules from mammalian cells⁶⁰. Similar surface localization was also observed in simulations of condensates formed by Arg-rich disordered proteins and polynucleotides.”*

2. Fig. 2a and text in the last paragraph on p. 6: The authors write “The dissolution of PLD-RLD droplets is preceded by a change in their color from yellow (RLD+PLD) to green (PLD), indication that RLD is leaving these droplets...”. Shouldn't it be ...from yellow (RLD+PLD) to red, indicating that PLD is leaving these droplets...?

We thank the reviewer for this point. Both statements are correct depending on the point of reference. The observation is that PLP-RRP (PLP for Prion-like polypeptide and RRP for Arginine-rich polypeptide) droplets in their initial stage are yellow (*green*-PLP+*red*-RRP), and in their final state they are green (PLP). In the intermediate transition. We have two sets of droplets, green droplets (PLP) and red (RRP+RNA) droplets. Thus, it depends on which of those droplets we connect to the initial state. We preferred to connect the PLP droplets to the initial state since the movie shows that the red droplets are born out of (or emerge from) the surface of the yellow droplets, so it is more likely that RRP is leaving. Especially that at higher PLP concentrations, it becomes clearer since what used to be yellow droplets turn to green without dissolution (see movie S2), and the red droplets (RRP-RNA) emerge from the surfaces of the previously yellow

droplets. Hence, we state that the yellow droplets turned green, indicating that RRP (the red component) leaves the condensates.

3. Fig. S3a: Why the size of PLD droplets alone is much larger than those in the presence of RLD? This needs to be clarified.

This is an excellent observation by the reviewer. The size difference is a consequence of two factors:

(a) RRP-PLP droplets form immediately upon mixing, where the nucleation of these condensates occurs throughout the sample. On the other hand, PLP droplets nucleate at the air-water interface. This difference leads to a higher number of coalescence events occurring for PLP droplets since they are continuously produced at the interface (earlier droplets fuse with later droplets because they are within proximity).

(b) The RRP-PLP droplets are charged (due to the presence of arginine residues), while PLP droplets are not. Hence there can be repulsive forces that slow down the rate of coalescence of RRP-PLP droplets (especially in the earlier stages of nucleation where droplets are small). These repulsive forces are absent in the case of homotypic PLP condensates. Also, the PLP-RRP droplets are likely to be more viscous (slower PLP mobility as probed by FRAP), which would negatively regulate their fusion kinetics and impede their growth in comparison to the homotypic PLP condensates.

Accordingly, we provided a discussion on this issue in the revised manuscript. The relevant sentences now read:

“We note that the difference in size between PLP droplets and PLP-RRP droplets may be a consequence of RRP-PLP condensates having excess charge due to the presence of Arg residues.”

4. Fig. 3c and text on p. 7: In contrast to what the authors state in the text, small RLD (red) PLD droplets appear to be also present without attachment to the surface of PLD (green) droplets. This should be clarified.

We thank the reviewer for pointing this out. We speculate that the formation of multiphasic condensates is a two-step process. First, the two types of condensates (PLP and RRP-RNA) form separately and undergo stochastic diffusion within the sample (see Movie S1). It is to be noted that the added RNA forms RRP-RNA droplets with RRP present in the dilute phase as well as by complexing with the RRP exiting the PLD-RRP condensates. Upon meeting, they form the multiphasic structure with a defined organization. In some cases, RRP-RNA condensates fall on the coverslip surface before meeting a PLP condensate, and their motion is arrested as they are attached to the surface. Therefore, they appear to exist as isolated droplets. We think this is the reason why one would observe isolated condensates of both kinds occasionally on the glass surface. However, in every case where condensates do meet, they form a distinct multiphasic morphology that is constant throughout the sample and reproducible over independent sample preparations. We have now added the following sentence in the main text to clarify this. *“This multiphasic pattern was persistent throughout the sample, although a few small RRP-RNA droplets were present as isolated droplets without interacting with PLP droplets.”*

5. p. 9, second line: It should be Fig. 4c (not (Fig. c)).

We apologize for this typographical error. We have fixed it in the revised manuscript.

REVIEWERS' COMMENTS

Reviewer #1 (Remarks to the Author):

I congratulate the authors. Their responses and revisions set a new standard for thoughtful engagement with reviewers and achieving clarity with rigor. As noted in the original review, this paper is going to stand the test of time and quickly become an influential citation classic. It was a thrill to read the revised version, and I suspect I will be reading this many times over. Nature Communications is fortunate to land this MS and I believe it is fully ready for publication with no further changes needed.

Reviewer #3 (Remarks to the Author):

The authors have adequately addressed my (relatively minor) concerns. In my opinion, this is a very interesting contribution of high significance to the field of protein LLPS.

Witold Surewicz